# Dual Ribosome Profiling reveals metabolic limitations of cancer and stromal cells in the tumor microenvironment

Daniela Aviles-Huerta[1,2,8], Rossella Del Pizzo[1,2,8], Alexander Kowar[1,2], Ali Hyder Baig[1], Giuliana Palazzo[1,2], Ekaterina Stepanova[1], Cinthia Claudia Amaya Ramirez[1], Sara D'Agostino[1], Edoardo Ratto[2,3], Catarina Pechincha[2,3], Nora Siefert[2,3], Helena Engel[2,4], Shangce Du[4,5], Silvia Cadenas-De Miguel[6], Beiping Miao[4], Victor M. Cruz-Vilchez[1], Karin Müller-Decker[7], Ilaria Elia[6], Chong Sun[4], Wilhelm Palm[3] & Fabricio Loayza-Puch[1]✉

The tumor microenvironment (TME) influences cancer cell metabolism and survival. However, how immune and stromal cells respond to metabolic stress in vivo, and how nutrient limitations affect therapy, remains poorly understood. Here, we introduce Dual Ribosome Profiling (DualRP) to simultaneously monitor translation and ribosome stalling in multiple tumor cell populations. DualRP reveals that cancer-fibroblast interactions trigger an inflammatory program that reduces amino acid shortages during glucose starvation. In immunocompetent mice, we show that serine and glycine are essential for optimal T cell function and that their deficiency impairs T cell fitness. Importantly, immune checkpoint blockade therapy imposes amino acid restrictions specifically in T cells, demonstrating that therapies create distinct metabolic demands across TME cell types. By mapping codon-resolved ribosome stalling in a cell-type-specific manner, DualRP uncovers metabolic crosstalk that shapes translational programs. DualRP thus offers a powerful, innovative approach for dissecting tumor cell metabolic interplay and guiding combined metabolic-immunotherapeutic strategies.

Cancers develop in heterogeneous tissue environments, relying on the tumor microenvironment (TME) to sustain growth, therapy resistance, and metabolic support[1–4]. In response to nutrient depletion, cancer cells seek additional resources to adapt to a metabolically challenging environment. However, how immune and stromal cell populations respond to nutrient depletion in the TME has been less explored. T cells, in particular, are susceptible to this hostile milieu[5–7]. Nutrient scarcity can severely impede T cell function and response to therapy[8]. While immune checkpoint blockade holds considerable promise, it frequently overlooks the metabolic hurdles that T cells and cancer cells encounter within the TME. Identifying such restrictions in different cell compartments within tumors remains challenging.

[1]Translational Control and Metabolism, German Cancer Research Center (DKFZ), Im Neuenheimer Feld 280, Heidelberg, Germany. [2]Faculty of Biosciences, University of Heidelberg, Heidelberg, Germany. [3]Division of Cell Signaling and Metabolism, German Cancer Research Center (DKFZ), Im Neuenheimer Feld 280, Heidelberg, Germany. [4]Immune Regulation in Cancer, German Cancer Research Center (DKFZ), Im Neuenheimer Feld 280, Heidelberg, Germany. [5]Faculty of Medicine, University of Heidelberg, Heidelberg, Germany. [6]Department of Cellular and Molecular Medicine, KU Leuven, 3000 Leuven, Belgium. [7]Core Facility Tumor Models, German Cancer Research Center (DKFZ), Im Neuenheimer Feld 280, Heidelberg, Germany. [8]These authors contributed equally: Daniela Aviles-Huerta, Rossella Del Pizzo. ✉e-mail: f.loayza-puch@dkfz-heidelberg.de

Ribosome profiling has emerged as a tool for detecting the availability of amino acids for protein synthesis[9–12]. This approach utilizes global ribosome occupancy to identify limiting amino acids. In principle, critical reduction of intracellular amino acid concentrations leads to tRNA deaminoacylation, resulting in ribosome stalling at a particular codon, which is indicative of limitations in the corresponding amino acid. Here, we introduce Dual Ribosome Profiling (DualRP), a method that enables the simultaneous study of translational programs and amino acid restrictions in distinct cell populations, both in vitro and in vivo.

Dual Ribosome Profiling (DualRP) combines cell-type-specific ribosome tagging with codon-resolved profiling to simultaneously measure translation and amino acid availability in interacting cell populations. We apply DualRP to reveal that cancer-associated fibroblasts rescue tumor protein synthesis under glucose deprivation by boosting lysosomal amino acid recycling and that immune checkpoint blockade creates serine and glycine shortages specifically in tumor-infiltrating T cells, impairing their cytotoxic activity. These findings uncover distinct metabolic vulnerabilities in cancer and immune compartments. DualRP therefore provides a versatile platform for mapping metabolic crosstalk in the tumor microenvironment and guides the design of combined metabolic and immunotherapeutic strategies.

## Results

### DualRP enables the study of ribosome occupancy in two interacting cell types

To study metabolic limitations in complex cell populations, we developed DualRP, a technique in which ribosomes from two separate cell populations are labeled with distinct chimeric proteins (GFP-RPL10a, mCherry-RPL10a, or NeonGreen-RPL10a)[13]. This labeling facilitates cell type-specific immunoprecipitation of ribosomes within a mixed population, followed by ribosome profiling (Fig. 1a). To accomplish this in the triple-negative breast cancer (TNBC) cell line SUM-159PT, we introduced N-terminal tags of GFP, mCherry, or mNeonGreen into the *RPL10a* gene through gene editing, achieving homozygous expression (Supplementary Fig 1a–d). Importantly, we verified that chimeric proteins were effectively incorporated into translating ribosomes without adversely affecting cell proliferation or global rates of protein synthesis (Supplementary Fig. 1e–g).

To assess the effectiveness and quality of the DualRP system, we first generated ribosome profiling libraries from pull-downs of tagged cells and compared them to libraries prepared using conventional sucrose gradients[14]. We observed a strong correlation in ribosome density in all comparisons (Supplementary Fig. 2a, b). Next, we co-cultured SUM-159PT-GFP-RPL10a cells and MRC5-mCherry-RPL10a fibroblasts and performed ribosome immunoprecipitation for each population. Immunoprecipitation assays and principal component analysis (PCA) of RPFs confirmed the specificity of the DualRP approach (Fig. 1b, c). RPFs originating from tagged ribosomes predominantly map to coding sequences (CDS). These fragments displayed a distinct 3-nt periodicity and exhibited a heightened density at the start codon (Supplementary Fig. 2c–f). Our results indicate that DualRP enables highly consistent and high-quality ribosome occupancy measurements in two interacting populations.

Amino acid starvation results in deaminoacylation of the corresponding tRNAs, leading to reduced interactions with site A of the ribosome[15,16]. DualRP can also be used to assess the interactions between tRNAs and ribosomes (Supplementary Fig. 2g). Through the pull-down of tagged ribosomes in co-cultures of breast cancer cells

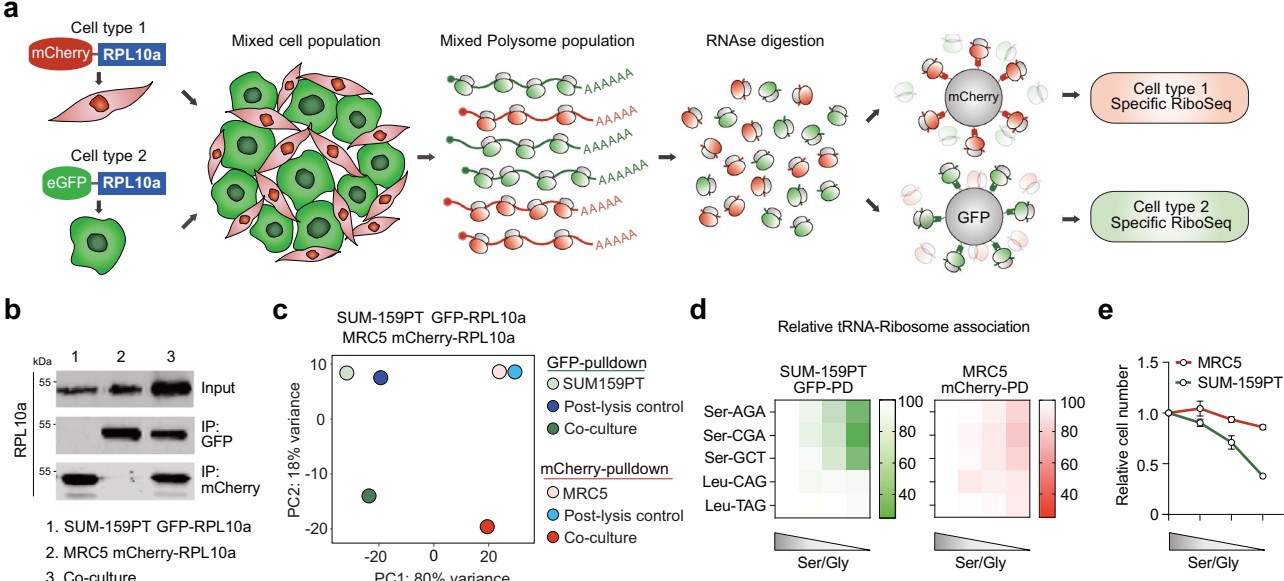

**Fig. 1 | DualRP allows for the study of ribosome occupancy and tRNA-Ribosome association in two interacting cell types. a** Schematic diagram of dual ribosome profiling (Dual-RP): Ribosomes from two distinct cell types are individually tagged with chimera proteins, eGFP-RPL10a and mCherry-RPL10a. The two cell populations are mixed, lysed, and ribosomes from each cell type are then immunoprecipitated using highly specific nanobodies. Subsequently, libraries for next-generation sequencing are prepared from the recovered ribosome-protected fragments (RPFs). **b** Immunoprecipitation experiments with beads coated with nanobodies against GFP (IP: GFP) or against mCherry (IP: mCherry), followed by western blot analysis in SUM-159PT-GFP-RPL10a and MRC5-mCherry-RPL10a cells. The blots are representative of 3 independent experiments. **c** Principal Component Analysis (PCA) from ribosome profiling libraries generated from pull-down of mono- and co-cultures of SUM-159PT-GFP-RPL10a and MRC5-mCherry-RPL10a. Mixes of lysed mono-cultures were used as controls (Post-lysis control). **d** tRNA–ribosome association assay in SUM-159PT-GFP-RPL10a and MRC5-mCherry-RPL10a cells cultured under serine/glycine starvation. Shading within the graph indicates decreasing serine/glycine concentrations, ranging from 0.4 mM to 0 mM. PD pull-down. Data are presented as mean ± SD (*n* = 3). **e** Normalized cell numbers at different serine/glycine concentrations. The data are expressed as relative cell numbers compared to the untreated group at the endpoint. Measurements were taken 48 hours after plating. Data represent mean ± SD from biologically independent experiments (*n* = 3).

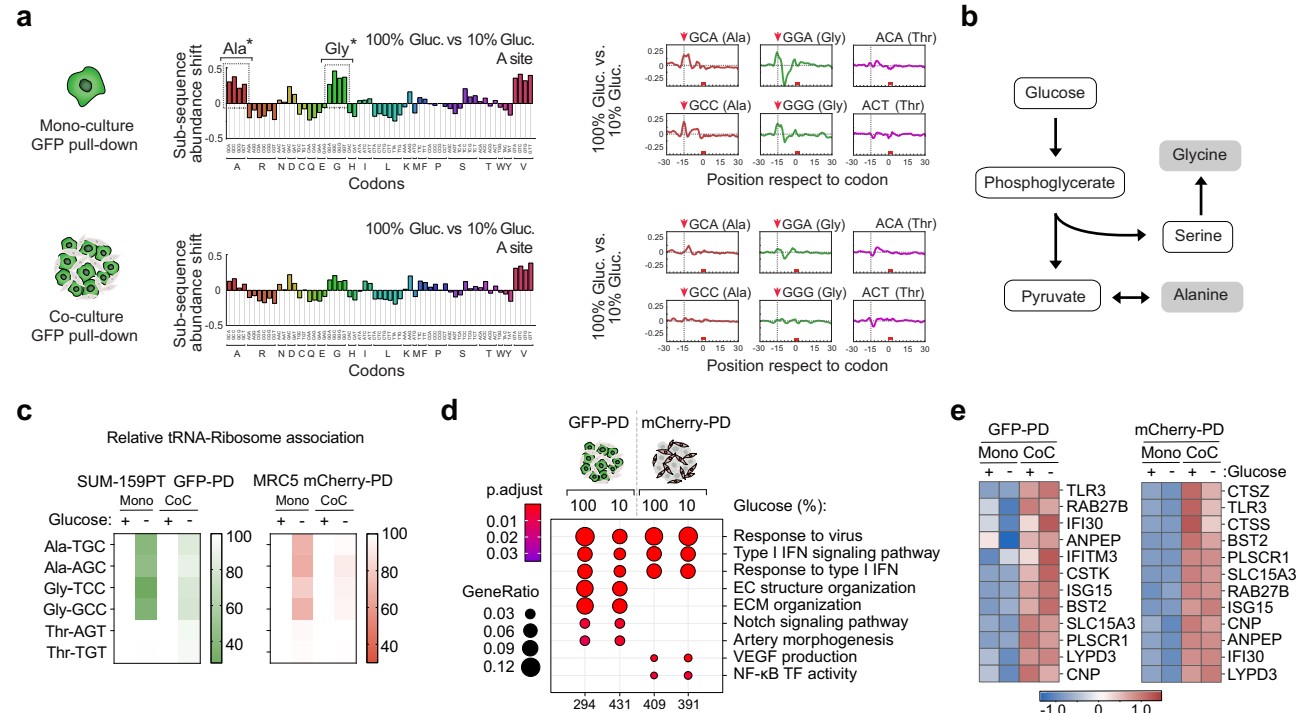

**Fig. 2 | DualRP reveals that heterotypic cell interactions restore ribosome stalling and tRNA aminoacylation rates during glucose starvation. a** Diricore analysis of GFP pull-downs from mono-cultures of SUM-159PT GFP-RPL10a cells (upper panel) and co-cultures with MRC5 mCherry-RPL10a (lower panel) grown in full (100%, 25 mM) or glucose-deprived (10%, 2,5 mM) medium. Cells were treated for 48 hrs. *Out-of-frame analysis. *p*-values were determined using a two-tailed z-test. *p* < 0.01 for the GCA, GCC, GCG, GCT Ala codons and for the GGA, GGC, GGG, GGT Gly codons. Source data including exact *p*-values are provided as Source Data file. **b** Schematic diagram of the synthesis of the amino acids alanine and glycine from glucose. **c** tRNA-Ribosome association assay in mono- and co-cultures (CoC) of SUM-159PT-GFP-RPL10a and MRC5-mCherry-RPL10a growing in full ((+), 25 mM) or glucose-deprived ((-), 2,5 mM) medium. Data represent mean ± SD (*n* = 3). PD pull-down. **d** Gene ontology analysis of genes upregulated upon heterotypic interaction of SUM-159PT GFP-RPL10a and MRC5 mCherry-RPL10a growing in full (100%, 25 mM) or glucose-deprived (10%, 2,5 mM) medium. Significantly enriched gene sets (*p*-value < 0.05) are shown, along with the number of identified proteins in the respective gene set (GeneRatio). Raw *p*-values come from a hypergeometric/ Fisher test. Adjusted *p*-values are then computed by the Benjamini-Hochberg procedure. PD pull-down. **e** Heatmaps displaying differential expression of representative lysosomal genes based on Ribo-seq counts in mono- and co-cultures of SUM-159PT GFP-RPL10a and MRC5 mCherry-RPL10a grown in full ((+), 25 mM) or glucose-deprived ((-), 2,5 mM) medium. LogFC is shown; PD pull-down.

and fibroblasts, we observed that cancer cell ribosomes exhibited decreased association with tRNAs when subjected to limiting concentrations of glutamine or serine/glycine (Fig. 1d; Supplementary Fig. 2h–j). Importantly, association rates in both cell populations were strongly correlated with cell growth (Fig. 1e; Supplementary Fig. 2h–j). In summary, DualRP offers the unique ability to simultaneously investigate ribosome occupancy and the association between ribosomes and tRNAs in two interacting populations.

### DualRP elucidates a type I IFN-mediated pathway that enhances amino acid availability for tRNA charging

To understand how different types of interacting cells respond to metabolically challenging conditions, we exposed co-cultures of breast cancer cells (SUM-159PT-GFP-RPL10a) and fibroblasts (MRC5-mCherry-RPL10a) to limiting concentrations of glucose. Using DualRP and differential ribosome codon reading (diricore) analysis[9], we discovered that cancer cells experienced ribosome stalling at alanine, glycine, and valine codons when grown individually as monocultures, indicating limitations in the corresponding amino acids (Fig. 2a). However, alanine and glycine deficiencies were remarkably alleviated when the cells were co-cultured (Fig. 2a). The same effect was observed in MDA-MB-231-GFP-RPL10a cells co-cultured with MRC5 fibroblasts (mCherry-RPL10a) and MDA-MB-231 cells carrying a different tag (NeonGreen-RPL10a; Supplementary Fig. 3a, d). Additionally, a similar response was noted in SUM-159PT cells where the *RPS3* gene was C-terminally tagged with GFP (RPS3-GFP; Supplementary Fig. 1h–l), indicating that ribosome stalling during glucose starvation is

independent of the cell line, protein tag, or ribosomal subunit tagging (Supplementary Fig. 3g). These findings suggest that the interplay between cancer cells and fibroblasts facilitate the exchange or synthesis of nutrients, providing a mechanism by which these cells collaboratively overcome the amino acid constraints imposed by glucose limitation.

Alanine and glycine can be generated de novo from glucose through pyruvate transamination and the serine synthesis pathway, respectively[17,18] (Fig. 2b). TNBC cells lacking PHGDH depend heavily on extracellular serine, which can be taken up from the environment or synthesized from glycine by serine hydroxymethyltransferase (SHMT)[19]. This glycine-to-serine conversion becomes crucial when glucose-derived de novo synthesis is impaired. Consistent with ribosome stalling, we observed reduced intracellular levels of alanine and glycine (Supplementary Fig. 3j) and a decrease in ribosome association with aminoacylated alanine (Ala) tRNAs (TGC and AGC) as well as glycine (Gly) tRNAs (TCC and GCC) following glucose starvation in monocultures of cancer cells and fibroblasts. By contrast, the association of aminoacylated control threonine (Thr) tRNAs (AGT and TGT) remained unchanged (Fig. 2c). However, in glucose-deprived co-cultures of SUM-159PT cells and MRC5 fibroblasts, the rate of ribosome association with Ala- and Gly-tRNAs was restored (Fig. 2c). A similar pattern was observed in MDA-MB-231, SUM-159PT-NeonGreen-RPL10a, and SUM-159PT-RPS3-GFP cells co-cultured with MRC5 fibroblasts (Supplementary Fig. 3b, e, h). Although reduced intracellular proline levels were also observed in SUM-159PT cells under glucose-limiting conditions (Supplementary Fig. 3j), we did not detect ribosome stalling

(Fig. 2a) or changes in proline tRNA aminoacylation, suggesting that this reduction is not sufficient to impact protein synthesis (Supplementary Fig. 3k). Collectively, these findings confirm the effectiveness of the DualRP system in investigating cell-type-specific amino acid deficiencies in cultured cells and reveal that heterotypic cell interactions restore tRNA aminoacylation rates during glucose starvation.

To explore the mechanism behind the restoration of tRNA aminoacylation in metabolically challenged co-cultures, we examined differences in Ribo-Seq-based expression signatures between mono- and co-cultures of breast cancer cells and fibroblasts in both full media and under limiting concentrations of glucose. Gene ontology (GO) and gene set enrichment analysis (GSEA) highlighted the significant enrichment of terms related to the type I IFN signaling pathway in both cell lines (Fig. 2d). Specific terms in SUM-159PT cells were associated with extracellular matrix (ECM) organization and the Notch signaling pathway, while MRC5 cells in co-culture upregulated genes involved in VEGF production and NF-κB transcription factor activity (Fig. 2d). These expression patterns were also observed in co-cultures grown with limiting concentrations of glucose in different cell lines and with different ribosomal tags (Supplementary Fig. 3c, f, i).

Among metabolic gene subsets, we observed a robust increase in the expression of genes localized in the lysosomes of both breast cancer cells and MRC5 fibroblasts (Fig. 2e; Supplementary Fig. 4a–e). The promoters of the upregulated lysosomal genes were found to be enriched in STAT1 binding sites, indicating their potential as interferon-stimulated genes (ISGs) (Supplementary Fig. 4f). As these co-cultures induced the IFN-I signaling pathway, activating JAK-STAT signaling and the expression of ISGs[20], this observation suggested that the subset of lysosomal genes might directly respond to IFN-I signaling pathway activation, thereby enhancing lysosomal nutrient generation.

To investigate this possibility, we measured lysosomal proteolysis using DQ BSA, a self-quenched albumin probe whose fluorescence becomes dequenched upon lysosomal proteolysis[21] (Fig. 3a). Coculturing resulted in a significant increase in intracellular DQ BSA fluorescence in both breast cancer cells and fibroblasts (Fig. 3b), indicating enhanced lysosomal catabolism. To determine whether this response was mediated by the IFN-I signaling pathway or glucose starvation, we treated mono-cultures of SUM-159PT cells with either IFN-β or limiting concentrations of glucose. Glucose depletion alone did not enhance lysosomal catabolism, whereas IFN-β treatment increased DQ BSA fluorescence both in full medium and under glucose-limiting conditions (Supplementary Fig. 4g). Next, we generated STAT1 knockout SUM-159PT cells (Supplementary Fig. 4h). Upon IFN-β treatment, these knockout cells failed to upregulate canonical ISGs and genes associated with lysosomal function (Supplementary Fig. 4i). Furthermore, upon co-culture, their intracellular DQ BSA fluorescence did not increase, either in full medium (Fig. 3c, d) or under glucose starvation (Supplementary Fig. 4j). STAT1 knockout cells were also unable to rescue the ribosome association of Ala- and Gly-tRNAs in glucose-deprived co-cultures (Fig. 3e). Thus, the IFN-I signaling pathway is required to promote lysosomal function and amino acid production in co-cultures exposed to limiting nutrient concentrations.

Lysosomal degradation of extracellular proteins serves as an amino acid source that cancer cells exploit to thrive under nutrient-poor conditions[22,23]. To establish whether lysosomal catabolism is necessary for producing charged tRNAs in glucose-starved co-cultures, we generated breast cancer cells deficient for LYSET (Supplementary Fig. 4k), which was recently identified to be required for lysosomal catabolism of macropinocytic and autophagic cargoes[24]. LYSET KO cells displayed reduced enzymatic activities of various lysosomal proteases but exhibited induction of ISGs to the same extent as control cells upon IFN-β treatment (Supplementary Fig. 4k–l). DQ BSA fluorescence and ribosome association of Ala- and Gly-tRNAs were strongly decreased in LYSET-deficient cells upon co-

culture with MRC5 cells (Fig. 3f–h; Supplementary Fig. 4m). Thus, the interaction between breast cancer cells and stromal cells enhances lysosomal catabolism, ultimately leading to greater nutrient availability for tRNA aminoacylation.

## DualRP detects specific metabolic limitations in different cell compartments of the TME

Having established the DualRP system in cells in culture, we sought to identify metabolic constraints simultaneously in distinct cell types within tumors. For this purpose, we adapted DualRP for studying amino acid restrictions in the cancer and T cell compartments of the TME. We crossed mice expressing the hemagglutinin (HA)-tagged ribosomal protein RPL22 (RiboTag mice) with CD4-Cre mice to selectively tag ribosomes in T cells with HA (CD4-Cre:RiboTag mice; Supplementary Fig. 5a)[25]. Simultaneously, we engineered a mouse tumor cell line (E0771) derived from spontaneous breast cancer in C57BL/6 mice, where the endogenous *Rpl10a* gene was homozygously tagged at the N-terminus with GFP (Supplementary Fig. 5b–d). We confirmed that GFP-RPL10a-expressing cells did not show proliferative or translational defects (Supplementary Fig. 5e–g). Western blot analysis of proteins immunoprecipitated with anti-HA from CD4-Cre:RiboTag homozygous mouse spleens demonstrated that RPL22-HA was exclusively detectable in mouse T cells and co-immunoprecipitated with other ribosomal proteins (RPL10a). By contrast, anti-HA did not pull-down ribosomal proteins from E0771-GFP-RPL10a cells (Supplementary Fig. 5h), confirming the specificity of our approach.

To enable in vivo interactions between tumor cells and host-derived T cells, we orthotopically injected CD4-Cre:RiboTag mice with E0771-GFP-RPL10a cells (Supplementary Fig. 6a). After tumors reached a defined size and recruited T cells, we collected tumor samples, flash-frozen them to capture native gene expression patterns and ribosome positions, and subsequently conducted DualRP. RPFs derived from both cancer and T cells predominantly mapped to coding sequences (CDS), exhibiting a clear 3-nt periodicity and displaying an increased density at the start codon (Supplementary Fig. 5i–m). Principal component analysis (PCA) of RPFs confirmed the specificity of DualRP in tumors (Supplementary Fig. 5n). Thus, DualRP can be used to study ribosome occupancy in multiple tumor cell compartments with high specificity.

To identify metabolic constraints within the cancer and T cell compartments in response to immune checkpoint blockade therapy, we administered anti-PD1 (αPD1) to CD4-Cre:RiboTag tumor-bearing mice and conducted DualRP analysis (Fig. 4a). Ribosome occupancy analysis revealed ribosome stalling at alanine, aspartic acid, serine, and glycine codons within the T cell compartment, while cancer cells showed no enrichment in any specific codons (Fig. 4b; Supplementary Fig. 6b). These findings suggest that these amino acids may be required for an effective T cell cytotoxic response.

Given that serine and glycine can be interconverted, and serine is particularly critical for T cell activation[26], we focused on these two amino acids. To further validate these observations, we used a system to induce and analyze tumor-specific T cell responses. Naive T cell receptor (TCR) transgenic OT-1 CD8 + T cells, which recognize the chicken ovalbumin (OVA)$_{257-264}$ peptide, were activated for 72 hours using anti-CD3, anti-CD28, and IL-2. These activated OT-1 cells were then co-cultured with E0771 breast cancer or TC1 lung adenocarcinoma cells constitutively expressing the OVA protein, either in complete medium or in medium deprived of serine and glycine. Co-culture in serine/glycine-deprived conditions ((-) SG) significantly reduced CD8 + T cell cytolytic activity, whereas increasing serine concentrations enhanced the response (Fig. 4c). Additionally, serine/glycine starvation decreased the expression of interferon γ (IFNγ) and granzyme B (GzmB) in OT-1 T cell co-cultures with multiple OVA-expressing cancer cell lines (Fig. 4d; Supplementary Fig. 6d, e).

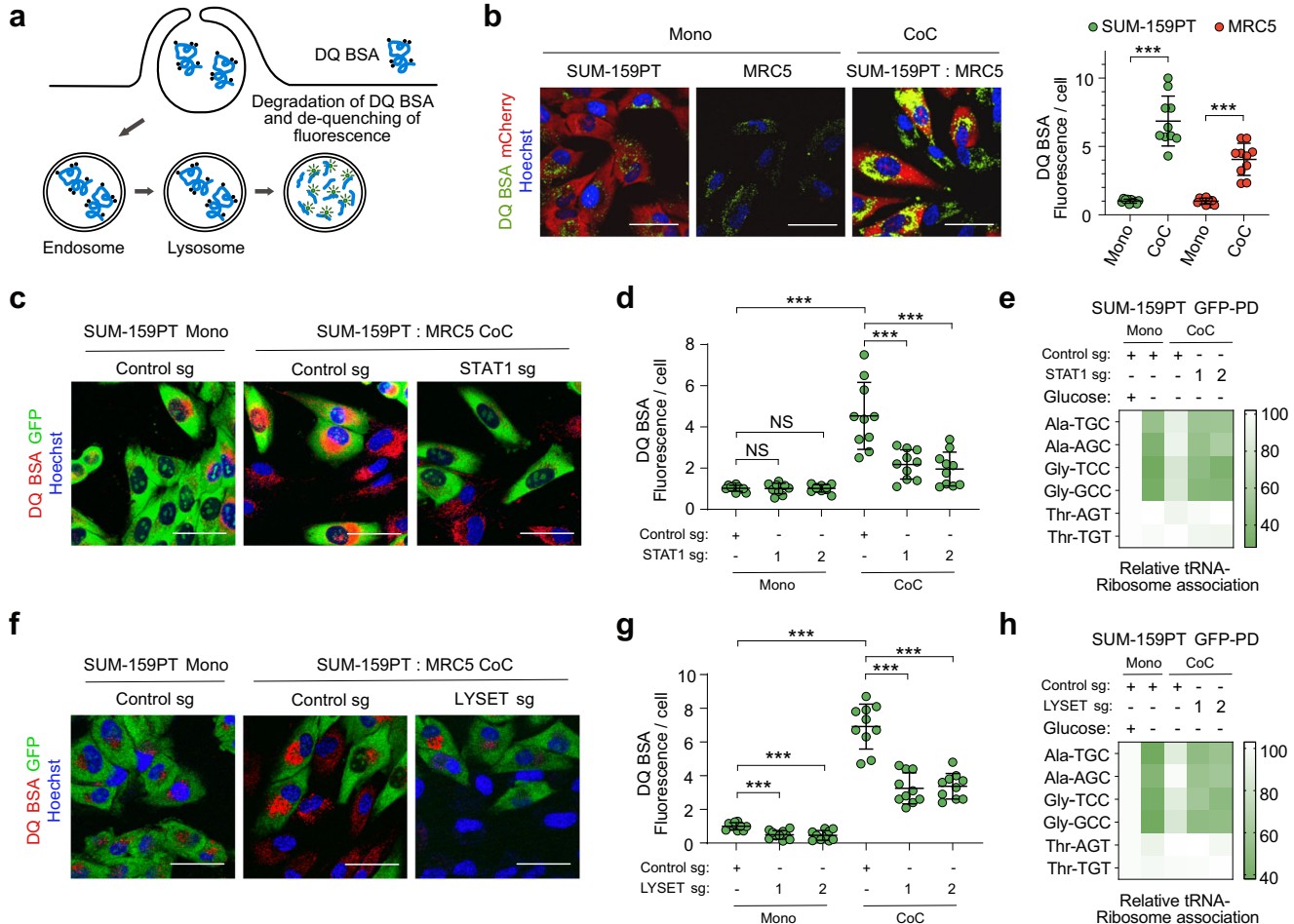

**Fig. 3 | Lysosomal catabolism is enhanced upon heterotypic interactions in a type I IFN-dependent manner. a** Schematic showing endo-lysosome formation and DQ BSA degradation. DQ BSA undergoes cellular endocytosis and transits from early endosomes to late endosomes. These late endosomes subsequently merge with lysosomes rich in acidic proteases. As a result, endo-lysosomes form, facilitating the degradation of DQ BSA and consequently restoring the fluorescence of the dye associated with this cargo. **b** Representative fluorescence microscopy images showing the degradation of lysosomal DQ BSA in mono- and co-cultures (CoC) of SUM-159PT-mCherry-RPL10a and MRC5 cells. Scale bars = 50 μm. Quantification of DQ BSA fluorescence in the indicated conditions (Right panel). Data represent mean ± SD from biologically independent experiments (n = 10); p-values were calculated using a two-tailed unpaired t-test. ***p < 0.001. Source data including exact p-values are provided as Source Data file. **c** Representative confocal microscopy images illustrating the degradation of lysosomal DQ BSA in SUM-159PT-GFP-RPL10a expressing sgRNAs targeting the STAT1 and growing as mono- or co-cultures (CoC) with MRC5. Scale bars = 50 μm. **d** Quantification of DQ BSA fluorescence in SUM-159PT GFP-RPL10a transduced with sgRNAs targeting the STAT1 gene or a control sequence growing as mono- or co-cultures (CoC) with MRC5 cells. The cells were maintained in full medium (Glucose 25 mM). Data

represent mean ± SD from biologically independent experiments (n = 10); p-values were calculated using a two-tailed unpaired t-test. NS non-significant; ***p < 0.001. Source data including exact p-values are provided as Source Data file. **e** Quantification of tRNA-Ribosome association in SUM-159PT-GFP-RPL10a expressing sgRNAs targeting STAT1 grown as mono- or co-cultures (CoC) with MRC5 cells in full ((+), 25 mM) or glucose-deprived ((-), 2,5 mM) medium. Data represent mean ± SD (n = 3). PD, pull-down. **f** Representative confocal microscopy images showing the degradation of lysosomal DQ BSA in SUM-159PT GFP-RPL10a expressing sgRNAs against LYSET and growing as mono- or co-cultures (CoC) with MRC5 cells. Scale bars = 50 μm. **g** DQ BSA quantification in SUM-159PT GFP-RPL10a transduced with sgRNAs targeting the LYSET gene growing as mono- or co-cultures (CoC) with MRC5 cells. Cells were grown in full medium (Glucose 25 mM). Data represent mean ± SD from biologically independent experiments (n = 10); p-values were calculated using a two-tailed unpaired t-test. ***p < 0.001. Source data including exact p-values are provided as Source Data file. **h** tRNA-Ribosome association assay in SUM-159PT-GFP-RPL10a expressing sgRNAs against LYSET growing as mono- or co-cultures with MRC5 cells in in full ((+), 25 mM) or glucose-deprived ((-), 2,5 mM) medium. Data represent mean ± SD (n = 3). PD pull-down.

These results were consistent in human cells. We co-cultured MDA-MB-231 cancer cells loaded with the melanoma antigen recognized by T cells 1 (MART-1) with T cells engineered to express a MART-1-specific T cell receptor (Supplementary Fig. 6f)[27]. In this model, serine/glycine deprivation significantly reduced tumor-reactive T cell cytotoxic activity and the expression of key T cell effector cytokines, including IL-2, TNF, and IFNγ (Supplementary Fig. 6g, h).

Next, to investigate the dynamics of serine metabolism in co-cultures of cancer and T-cells during antigen-specific responses, we employed a cell-size filtration-based technique enabling the rapid separation of both populations (Supplementary Fig. 6c)[28]. We treated

mono- and co-cultures of OT-1 CD8 + T cells and E0771-OVA cells with U-[$^{13}$C]-glucose, and quantified the incorporation of $^{13}$C-glucose-derived carbon into serine and glycine using liquid chromatography-mass spectrometry (LC-MS). Intracellular pools of serine and glycine in naïve T cells incorporated a minimal fraction of glucose into serine and glycine; however, approximately 30% of the intracellular pools of serine and glycine were labeled from glucose upon antigen-specific activation (Fig. 4e, f). Conversely, glucose-dependent labeling of serine and glycine in E0771-OVA cells slightly decreased upon heterotypic interaction with OT-1 CD8+ T cells (Fig. 4e, f), suggesting that upon antigen recognition, T cells activated serine and glycine synthesis from

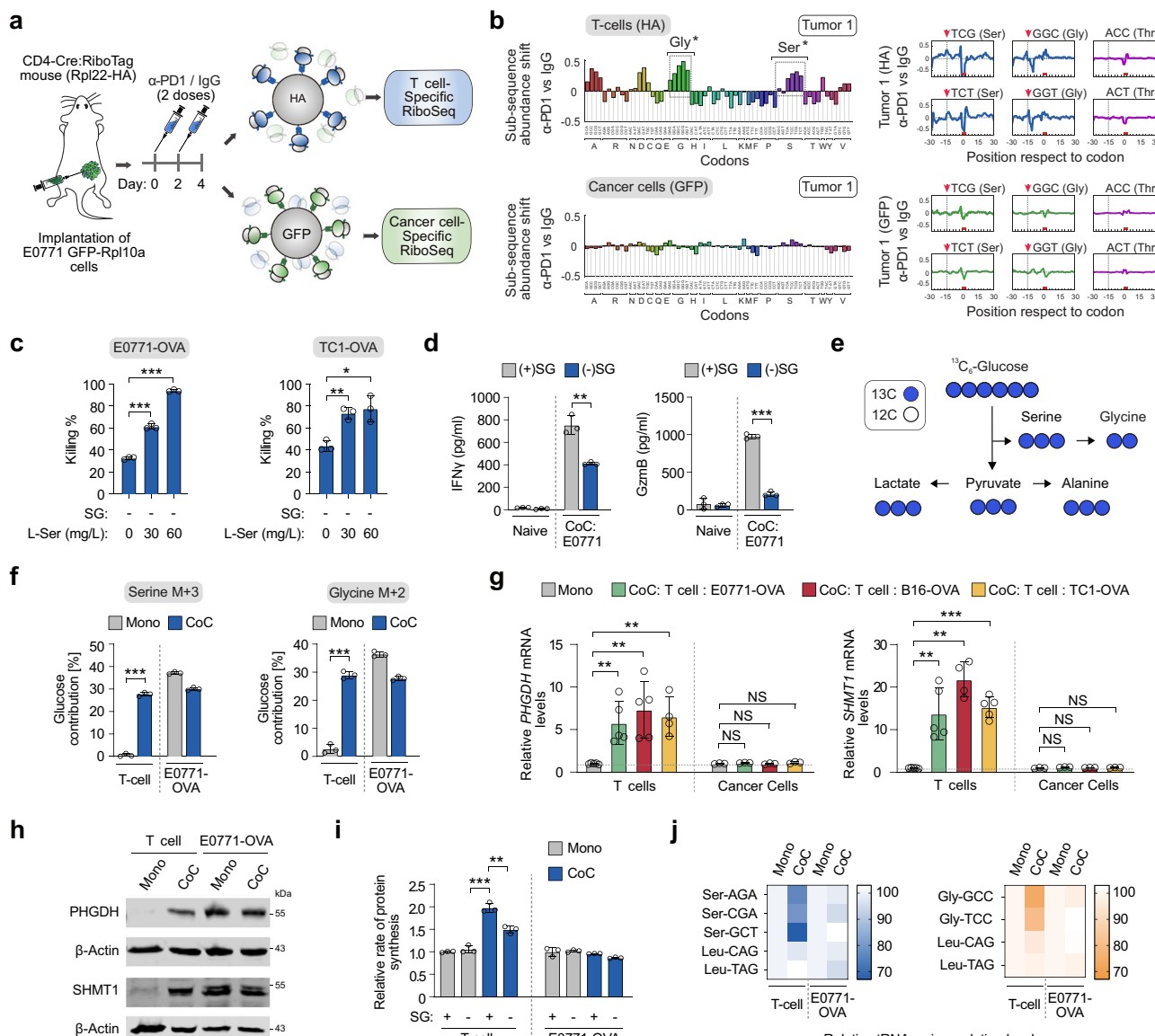

**Fig. 4 | Serine and glycine are limiting amino acids in the T cell compartment of tumors and are necessary for an effective T cell cytotoxic response. a** Schematic of experimental design. Female CD4-Cre:RiboTag mice (7-8 weeks-old) were treated with two doses of anti-PD1(2 μg/μl) or IgG (2 μg/μl). Two days after the last dose, tumors were flash frozen, lysed, and ribosomes from each cell type were immunoprecipitated using either anti-HA or anti-GFPs coated beads. **b** Diricore analysis of tumors treated as described in (**a**). Ribosome stalling in tumor-infiltrated T cells and E0771 tumor cells are shown in the upper and lower panels, respectively. Ribosome density at the A site is shown. *Out-of-frame analysis. *p*-values were determined using a two-tailed z-test. *p* < 0.01 for the TCA, TCC, TCG, TCT Ser codons and for the GGA, GGC, GGG, GGT Gly codons. Source data including exact *p*-values are provided as Source Data file. **c** Killing assay by CD8 + T cells in co-culture with E0771-OVA or TC1-OVA cells in serine/glycine (SG)-deprived medium. Data represent mean ± SD from biologically independent experiments (*n* = 3); *p*-values were calculated using a two-tailed unpaired *t*-test. **p* < 0.05; ***p* < 0.01; ****p* < 0.001. Source data including exact *p*-values are provided as Source Data file. **d** Quantification of IFNγ and GzmB levels produced by CD8 + T cells when co-culture with E0771-OVA cancer cells in full medium or SG-deprived medium. Data represent mean ± SD from biologically independent experiments (*n* = 3); *p*-values were calculated using a two-tailed unpaired *t*-test. ***p* < 0.01; ****p* < 0.001. Source data including exact *p*-values are provided as Source Data file. **e** Schematic for $^{13}C_6$-

Glucose tracing into the serine/glycine synthesis pathway. Blue circles represent $^{13}C$ atoms while white circles represent $^{12}C$ atoms. **f** Contribution of $^{13}C_6$-Glucose to serine M + 3 and glycine M + 2 in OT-1 CD8 + T cells and E0771-OVA breast cancer cells growing as mono- or co-cultures (CoC). Data represent mean ± SD from biologically independent experiments (*n* = 3); *p*-values were calculated using a two-tailed unpaired *t*-test. ****p* < 0.001. Source data including exact *p*-values are provided as Source Data file. qRT-PCR quantification (**g**) and Western blotting (**h**) of the indicated genes in co-cultures (CoC) of OT-1 CD8 + T cells and OVA-expressing cancer cell lines. Cells were separated by size after 24 hours of co-culture (Supplementary Fig. 6c). Data represent mean ± SD from biologically independent experiments (*n* = 5 for T cells and *n* = 3 for cancer cells); NS not significant; *p*-values were calculated using a two-tailed unpaired *t*-test. ***p* < 0.01; ****p* < 0.001. Source data including exact *p*-values are provided as Source Data file. **i** Protein synthesis rates based on OP-Puro incorporation in OT-1 CD8 + T cells and E0771-OVA breast cancer cells growing as mono- or co-cultures (CoC). Cells were grown in full medium (SG + ) or serine/glycine-deprived medium (SG−). Data represent mean ± SD from biologically independent experiments (*n* = 3); *p*-values were calculated using a two-tailed unpaired *t*-test. ***p* < 0.01; ****p* < 0.001. Source data including exact *p*-values are provided as Source Data file. **j** Ser-tRNAs, Gly-tRNAs and control Leu-tRNAs aminoacylation analysis in OT-1 CD8 + T cells and E0771-OVA breast cancer cells growing as mono- or co-cultures (CoC).

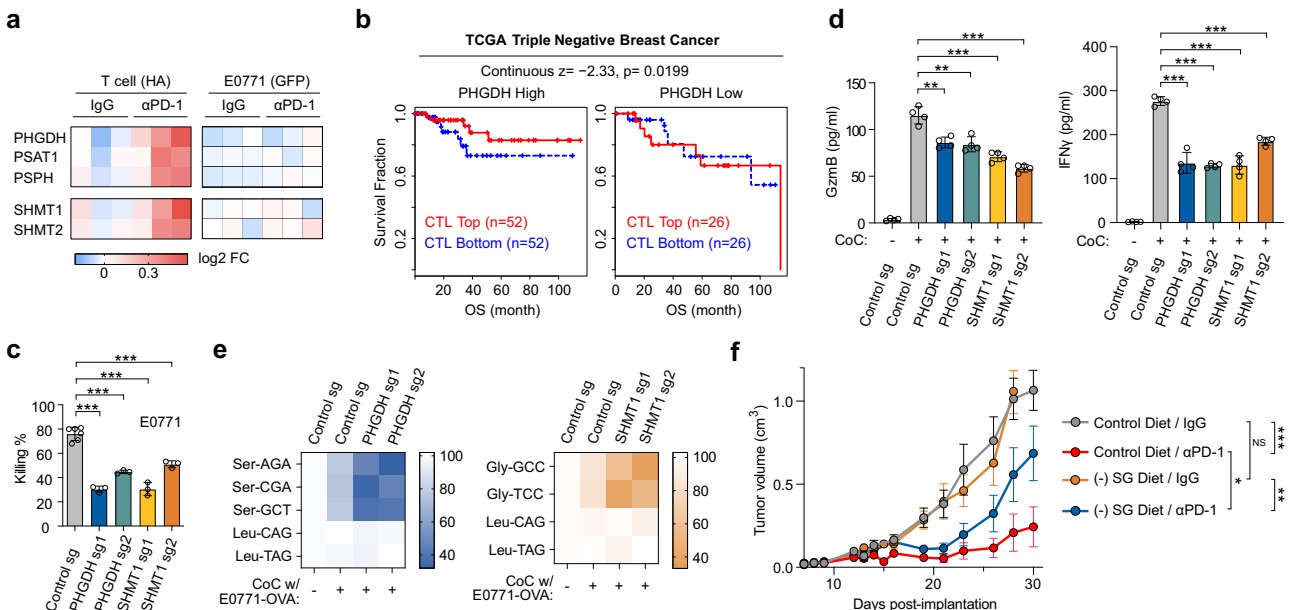

**Fig. 5 | Serine and glycine are crucial for effective immunotherapy, with the serine synthesis pathway associated with T cell dysfunction in cancer patients.** **a** Heatmap depicting the log2 fold-change (FC) values of differentially expressed genes in T cells and E0771 cancer cells from tumors treated wither with IgG (2 μg/μl) or αPD-1 (2 μg/μl). **b** TIDE analyses were conducted on PHGDH expression within T cell dysfunction signatures associated with improved survival in patients with triple negative breast cancer. TGCA, The Cancer Genome Atlas. *p*-values are calculated by two-sided Wilcoxon rank-sum test. OS, overall survival. **c** Cytotoxicity assay by CD8 + T cells expressing sgRNAs targeting PHGDH or SHMT1 and co-cultured with GP$_{33-41}$-pulsed E0771 cells for 24 hrs. Data represent mean ± SD from biologically independent experiments (*n* = 3); *p*-values were calculated using a two-tailed unpaired *t*-test. **p < 0.01. Source data including exact *p*-values are provided as Source Data file. **d** GzmB and IFNγ secretion by CD8 + T cells transduced with sgRNAs against PHGDH or SHMT1 and co-cultured with GP$_{33-41}$-pulsed E0771 cells for 24 hrs. Data represent mean ± SD from biologically independent experiments

(*n* = 4); *p*-values were calculated using a two-tailed unpaired *t*-test. **p < 0.01; ***p < 0.001. Source data including exact *p*-values are provided as Source Data file. **e** tRNA aminoacylation assay in T cells expressing sgRNAs targeting PHGDH or SHMT1 and co-cultured with GP$_{33-41}$-pulsed E0771 cells. Cells were separated by size (as described in Supplementary Fig. 6c) after 24 hours of co-culture. Ser-tRNAs, Gly-tRNAs and control Leu-tRNAs were analyzed. **f** Female C57BL/6 J mice (7–8 weeks old; *n* = 7 per group) were orthotopically injected with E0771 cells. Seven days later, mice were switched to either a control diet or a serine/glycine-free (− SG) diet. Once tumors became palpable, animals in each dietary group received intraperitoneal injections of control IgG or anti-PD-1 antibody (2 μg per dose). Tumor volumes are presented as the mean ± SD from biologically independent experiments (*n* = 7); *p*-values were calculated using a two-tailed unpaired *t*-test. NS non-significant; *p < 0.05; **p < 0.01; ***p < 0.001. Source data including exact *p*-values are provided as Source Data file.

glucose. These findings were consistent with the increased expression of genes associated with the serine biosynthesis pathway (*PHGDH*, *PSAT1*, and *PSPH*) and genes enabling serine entry into one-carbon metabolism (*SHMT1* and *SHMT2*), which were observed exclusively in OT-1 CD8 + T cells after physical interaction with OVA-expressing cancer cells or SIINFEKL-pulsed antigen presenting cells (Fig. 4g, h; Supplementary Fig. 7a–d).

Consistent with these findings, global rates of protein synthesis increased in OT-1 T cells following antigen recognition in a serine/glycine-dependent manner, while protein synthesis rates in E0771-OVA cells remained unchanged (Fig. 4i). Moreover, when measuring tRNA aminoacylation levels in co-cultures of cancer and T cell populations during antigen-specific responses (Supplementary Fig. 6c), we observed a significant decrease in the aminoacylation levels of serine (Ser)-tRNAs and glycine (Gly)-tRNAs in OT-1 T cells, whereas the levels remained unaffected in E0771-OVA cancer cells (Fig. 4j). These results were replicated in human MDA-MB-231 and MART-1 T cell co-cultures (Supplementary Fig. 6i, j). Collectively, our data demonstrate that serine and glycine are crucial for T cell protein synthesis in the TME within the context of immune checkpoint blockade and are required amino acids for an efficient T cell cytotoxic response.

### The availability of serine and glycine is a critical requirement for effective immune checkpoint blockade

Our in vivo DualRP datasets revealed an increase in the expression of genes linked to the serine biosynthesis pathway (*PHGDH*, *PSAT1*, and *PSPH*) and those enabling serine entry into one-carbon metabolism

(*SHMT1* and *SHMT2*) in the T cell compartment after the administration of αPD-1. By contrast, cancer cells did not exhibited changes (Fig. 5a), suggesting that the serine synthesis and the one carbon metabolism pathways are enhanced exclusively in the T cell compartment of tumors in response to checkpoint inhibition.

Next, to assess the clinical relevance of serine metabolism in cancer, we employed Tumor Immune Dysfunction and Exclusion (TIDE) analysis to investigate whether PHGDH expression in T cells correlates with outcomes across different cancer types[29]. Intriguingly, while a high cytotoxic T lymphocyte (CTL) score was associated with an overall survival benefit, low levels of PHGDH expression abolished this benefit in cases of TNBC, melanoma, lung adenocarcinoma, and endometrial tumors (Fig. 5b; Supplementary Fig. 8a–d), highlighting the importance of de novo serine synthesis in predicting cancer immunotherapy outcomes in patients.

Subsequently, we aimed to determine the essentiality of the serine synthesis and the one carbon metabolism pathways to ensure an effective cytotoxic T-cell response. To this end, we transduced Cas9-expressing CD8[+] T cells derived from P14 x CD4-Cre x Rosa-LSL-Cas9-GFP mice with sgRNAs targeting either PHGDH or SHMT1 (Supplementary Fig. 7e, f)[30]. These T cells express a lymphocytic choriomeningitis virus (LCMV) glycoprotein 33 (GP33)-specific T cell receptor (TCR). Remarkably, we observed that T cells lacking PHGDH or SHMT1 displayed a significant reduction in cytotoxic activity when co-cultured with GP$_{33-41}$ peptide-pulsed E0771, B16, or TC1 cells, in comparison to the control vector (Fig. 5c; Supplementary Fig. 7g). Furthermore, we studied the immunological traits of PHGDH and

SHMT1 KO CD8+ T cells. Interestingly, these mutants exhibited decreased levels of granzyme B and IFNγ when co-cultured with pulsed E0771 cells (Fig. 5d). Moreover, tRNA aminoacylation levels of Ser- and Gly-tRNAs in CD8+ T cells expressing PHGDH sgRNAs were significantly reduced upon antigen recognition (Fig. 5e). Our findings collectively demonstrate that genetic perturbations in the serine synthesis and one-carbon metabolism pathways lead to diminished cytotoxic activity in CD8+ T cells and a reduction in the availability of serine and glycine for protein synthesis.

To investigate the significance of serine and glycine in the effective response to checkpoint blockade immunotherapy, we employed a syngeneic breast cancer model with the murine cell line E0771. After injecting cancer cells into the mammary fad pads, we transitioned the mice to either a control diet or a serine/glycine-free ((-) SG) diet. When the tumors became visible, mice were treated with either αPD-1 or IgG. Notably, dietary intervention alone did not affect tumor growth (Fig. 5f). However, administering αPD-1 as monotherapy substantially reduced the tumor burden and markedly improved survival (Supplementary Fig. 8e). When combining the (-) SG diet with αPD-1 treatment, it considerably attenuated the effects of immune checkpoint blockade (Fig. 5f; Supplementary Fig. 8e). This reduction in efficacy correlated with changes in tumor-specific T-cell numbers within the tumors (Supplementary Fig. 8f), suggesting that systemic serine and glycine are key to an efficient immunotherapeutic response. In summary, these data demonstrate the utility of DualRP not only in concurrently analyzing gene expression in two interacting cell populations in vitro, but also in revealing the metabolic constraints within distinct cellular compartments of tumors. This tool has enabled us to establish that serine and glycine are crucial for effective immunotherapy and that the serine synthesis pathway is associated with T cell dysfunction-related clinical outcomes in cancer patients.

## Discussion
Various cell types within the tumor microenvironment (TME), including cancer-associated fibroblasts, endothelial cells, and immune cells, profoundly influence the metabolic behavior of cancer cells[31–35]. For example, interactions between stromal fibroblasts and triple-negative breast cancer cells induce the expression of ISGs[36–39], a phenotype associated with resistance to ionizing radiation[40,41]. Our DualRP analysis reveals that type I interferon (IFN) signaling activated during these heterotypic interactions remodels lysosomal function, enabling cancer cells to generate the nutrients necessary for tRNA aminoacylation, a finding that aligns with recent reports of IFN-enhanced lysosomal activity during bacterial infections[42].

Nutrient deficiencies within the TME can impair T cell activation and diminish their tumor-killing capacity. In vitro studies have shown that limitations in amino acids, such as serine, arginine, asparagine, and alanine, compromise T cell function[26,43–45], and tumors rich in metabolically exhausted T cells are often resistant to immune checkpoint inhibitors[46–48]. Using the DualRP system, we identified that T cells predominantly experience shortages in serine and glycine, which adversely affect their fitness and antitumor response. Intriguingly, following antigen recognition, T cells not only synthesize serine and glycine de novo from glucose but also may acquire extracellular serine, possibly exported by cancer cells. This dual sourcing expands our understanding of metabolic crosstalk within the TME and highlights the critical role of serine and glycine metabolism in supporting T cell function during immune checkpoint blockade.

Our analysis further indicates that immune checkpoint blockade therapy affects at least four serine codons, in contrast to the codon-specific reduction at TCC and TCT serine codons reported by Banh et al.[12]. This discrepancy likely stems from differences in experimental conditions, such as the duration of serine deprivation and the cell types analyzed, suggesting that codon-specific responses to serine depletion may vary with cellular context and nutrient availability.

Although both CTLA-4 and PD-1 suppress T cell activity, they do so via distinct mechanisms[49]. CTLA-4 acts early during T cell activation by competing with CD28 for co-stimulatory signals, thereby raising the activation threshold and dampening responses to weak antigens. In contrast, PD-1 functions later in the immune response by inhibiting signaling pathways such as PI3K/AKT within the TME[49,50]. While both receptors are linked with reduced glycolysis and lipid metabolism in T cells[51,52], their impacts on amino acid pathways, including serine synthesis, remain less well defined.

Recent studies demonstrated that formate or methanol supplementation can enhance effector T cell fitness and substantially improve the efficiency of immunotherapy[53,54]. Our findings suggest that serine depletion impairs not only nucleotide production and cellular proliferation[17,55] but also tRNA charging. Notably, tRNA charging can be rescued without fully restoring protein synthesis, potentially through amino acid-responsive pathways like GCN2[56], implying that ribosome stalling at specific codons (e.g., those for alanine or glycine) may modulate signaling rather than directly reduce protein output.

These insights support the potential of combinatorial treatments that pair checkpoint inhibitors with dietary restoration of one-carbon units and serine/glycine supplementation to enhance immunotherapy efficacy. Advances in metabolic profiling of distinct cell types have deepened our understanding of the TME's complexity[57–59], and our DualRP tool offers a versatile means to investigate metabolic constraints across different cellular populations.

Despite the strengths of our approach, challenges in achieving high-resolution ribosome profiling remain. Signal-to-noise issues in our diricore analysis, stemming from library preparation and limited starting material, necessitate cautious interpretation. Although our follow-up analyses, particularly those examining ribosome–tRNA associations, support the validity of our findings, future improvements in footprinting strategies and computational deconvolution will be crucial for enhancing codon-resolved reliability[14,60,61].

Finally, while our current focus is on amino acid limitations, DualRP's ability to capture ribosome footprints across mRNAs provides an opportunity to identify translationally regulated genes independent of transcriptional changes. Future studies may correlate ribosome occupancy with protein expression via proteomics or western blot analyses, thereby extending our approach to investigate multidimensional metabolic interactions in cancer and organismal development.

## Methods
### Cell culture
Human cell lines (SUM-159PT, MRC5, MDA-MB-231) were cultured in DMEM High Glucose Medium (Thermo Fisher; cat. 41966-029) supplemented with 10% FBS (Thermo Fisher; cat. 10270-106) and 100 units/ml penicillin and 100 mg/ml streptomycin (Thermo Fisher; cat. 15140-122). For glucose starvation experiments, cells were cultured for 48 h in DMEM without glucose and pyruvate (Thermo Fisher; cat. 11966-025) supplemented with either 4.5 g/l (high glucose) or 0.45 g/l glucose (low glucose), 10% dialyzed fetal bovine serum (Thermo Fisher; cat. 2093867) and 100 units/ml penicillin and 100 mg/ml streptomycin.

Mouse cell lines (B16, E0771 and TC1) were grown in RPM1 1640 (Thermo Fisher; cat. 21875-034) supplemented with 10% FBS, 100 units/ml penicillin and 100 mg/ml streptomycin (Thermo Fisher; cat. 15140-122), 1 mM sodium pyruvate (Thermo Fisher; cat. 11360-070), 20 mM HEPES (Sigma; cat.H4034) and 50 μM β-mercaptoethanol (Sigma; cat. M6250). Mouse cell lines expressing chicken ovalbumin (B16-OVA and TC1-OVA) or the lymphocytic choriomeningitis virus (LCMV) glycoprotein 33 (B16-GP33) were kindly donated by Dr. Guoliang Cui and Dr. Chong Sun. Platinum-E cells were used at low passage and cultured in DMEM (Thermo Fisher; cat. 41966-029) supplemented

with 10% FBS (Thermo Fisher; cat. 10270-106), 100 units/ml penicillin and 100 mg/ml streptomycin (Thermo Fisher; cat. 15140-122) and 1% GlutaMAX (Thermo Fisher; cat. 35050061). All cell lines were regularly tested for Mycoplasma contamination and kept at 37 °C in 5% CO2.

## Homozygous knock-in of ribosomal fluorescent proteins

Endogenous fluorescent tagging of ribosomal proteins RPL10a and RPS3 was performed in triple-negative breast cancer lines via homozygous knock-in with paired CRISPR–Cas9 as described by Koch et al.[62]. In brief, cells were co-transfected with (1) a Cas9D10A-nickase/sgRNA plasmid (Addgene 42335) targeting the desired ribosomal-protein locus and (2) a donor plasmid encoding GFP- or mCherry-tagged RPL10a or RPS3 flanked by ~800 bp homology arms (see Supplementary Data 1). Forty-eight hours post-transfection, cells were collected, washed in PBS + 2% FBS, and filtered through a 35 µm mesh for flow-cytometric single-cell sorting on a BD FACSAria II. Sorting gates were applied sequentially as follows: (1) FSC-A vs SSC-A to exclude debris; (2) FSC-A vs FSC-H to select singlets; (3) exclusion of dead cells using a live/dead dye; (4) fluorophore (GFP or mCherry) positivity, with gates set against untransfected parental controls; and (5) a high-fluorescence gate to enrich for putative homozygous integrants. Single cells from the final gate were deposited into 96-well plates. Compensation and gate boundaries were defined each day using unstained, single-color, and fluorescence-minus-one controls. Clones expanding from single cells were genotyped by PCR to confirm homozygous knock-in and tested for polysome incorporation of the fluorescent chimera. Clonal proliferation over 8 days was measured on an IncuCyte S3 Live-Cell Analysis System and compared to parental controls.

## Lentiviral overexpression and knock out generation

MRC5 fibroblasts expressing heterozygous ribosomal tagging (MRC5-mCherry-RPL10a) were produced via lentiviral overexpression. Virus was generated in HEK293Tx using PEI max transfection reagent (Polyscience; cat. 24765-1), lentiviral packaging plasmids (Addgene: 12251, 12253 and 14888) and the vector coding for the fluorescent chimera (pCDH1-mCherry-RPL10a). After viral transduction in presence of polybrene (Santa Cruz; cat. Sc-255611), cells were sorted for mCherry via FACS. sgRNA against STAT1 and LYSET were cloned into a lentiviral expression vector for Cas9 (plentiCRISPR v2; addgene 52961). SUM159PT and MRC5 knock outs were prepared via lentiviral transduction and selected with 2 µg/ml puromycin (Sigma; cat. P8833) for 5 days. E0771 cells expressing chicken ovalbumin (E0771-OVA) were generated via lentiviral transduction as described above and selected with 10 µg/ml blasticidin (Santa Cruz; cat.sc-495389) for 3 days.

## Immunoblotting

Cell extracts were separated on 12% SDS–PAGE gels and transferred to 0.45 µm nitrocellulose membranes (Biorad; cat. 1620115). Antibodies used were RPL10a (Abcam; cat. Ab174318, 1:1,000), RPS3 (Cell Signaling; cat. 9538, 1:1,000), eGFP (Santa Cruz; cat. sc-9996, 1:1,000), mCherry (Abcam; cat. Ab167453 1:1,000), Neongreen (Chromotek; cat. 32F6, 1:1,000), RPL22 (Santa Cruz; cat. sc-373993 1:500), HA-tag (Biolegend; cat. 901501, 1:1000), STAT1 (Cell Signaling; cat. 9172, 1:1,000), PHGDH (Sigma; cat. HPA021241, 1:1,000), SHMT1 (Cell Signaling;cat. 80715, 1:1,000), LYSET (Atlas Antibodies; cat. HPA048559, 1:1,000), Cathepsin B (R&D Systems; cat. AF953, 1:1000), Cathepsin L (R&D Systems; cat. AF952 1:1000), HEXA (R&D Systems; cat. AF6237, 1:1000).

For immunoblotting of polysome fractions, cells were washed with ice-cold PBS with 100 µg/ml cycloheximide (CHX) and lysed in NP40 lysis buffer (20 mM Tris-HCl pH. 7.5 pH, 10 mM MgCl2, 150 mM KCl, 1% NP40, 2 mM DTT, 1x EDTA-free Complete protease inhibitors and 100 µg/ml CHX). Cell extract was pipetted on top of a sucrose gradient and centrifuged for 2 hours at 36.000 rpm and 4 °C in the

Optima XPN 100 ultracentrifuge with swinging the bucket rotor SW41 Ti (Beckman Coulter). Fractions were loaded in 12% SDS–PAGE gels and transferred as described above.

## Dual ribosome profiling

SUM-159PT–GFP-RPL10a and MRC5–mCherry-RPL10a cells were co-cultured at a 1:1 ratio for 48 h. On harvest day, cultures were washed with ice-cold PBS containing 100 µg/mL cycloheximide (CHX), scraped, and pelleted (1,300 × g, 5 min, 4 °C). Cell pellets were resuspended in 1 mL NP-40 lysis buffer (20 mM Tris-HCl pH 7.8, 10 mM MgCl2, 150 mM KCl, 1% NP-40, 2 mM DTT, 1× protease inhibitors, 100 µg/mL CHX) and incubated on ice for 15 min.

Tumor tissue was snap-frozen in liquid nitrogen immediately after excision, then homogenized on ice for 30 s at 2,500 rpm (Mikro Dismembrator-S) in the same NP-40 buffer. Lysates were clarified by centrifugation (1300 × g, 10 min, 4 °C), and the supernatant treated with RNase I (1 U/µL; Ambion; cat. AM2294) for 30 min at room temperature with rotation to generate monosomes.

Monosomes were immunoprecipitated for 1 h at room temperature using 40 µL of GFP-Trap, RFP-Trap, or NeonGreen-Trap magnetic beads (Chromotek), or Protein A/G beads (Thermo Fisher; cat. 88802) pre-loaded with anti-HA (Biolegend; cat. 901501). Beads were washed three times in NP-40 lysis buffer and three times in NP-40 wash buffer (20 mM Tris-HCl pH 7.8, 10 mM MgCl2, 350 mM KCl, 1% NP-40, 2 mM DTT, protease inhibitors, 100 µg/mL CHX).

To release ribosome-protected fragments (RPFs), beads were incubated 1 h at 45 °C in 300 µL NP-40 lysis buffer supplemented with 1% SDS and 5 µL proteinase K (Roche 3115828001). The supernatant was extracted with TriReagent (Zymo; cat. R2050-1).

Libraries were prepared according to McGlincy, N. J. & Ingolia, N. T.[14]. Briefly, RPFs (20–34 nt) were size-selected on a 12% denaturing polyacrylamide-urea gel, purified by ethanol precipitation, then dephosphorylated (T4 PNK; NEB cat. M0201S). A pre-adenylated DNA linker was ligated (T4 Rnl2(tr) K227Q; NEB cat. M0351S), and excess linker removed with 5′ Deadenylase and RecJf (NEB cat. M0331S, cat. M0264S). Ribosomal RNA was depleted using biotinylated rRNA oligos (Supplementary Data 1) and streptavidin beads (Thermo Fisher cat. 65001).

cDNA synthesis (SuperScript III; Thermo Fisher cat. 18080051) was followed by purification on an 8% polyacrylamide-urea gel. cDNA was circularized (CircLigase II; Biosearch cat. CL9021K), PCR-amplified (Q5; NEB cat. M0492S), cleaned (Zymo cat. D4013), size-selected on an 8% gel, and quantified by Qubit HS DNA assay. Libraries were normalized to 2 nM for sequencing.

## tRNA aminoacylation assay

Cells were harvested and resuspended in a solution of 0.3 M sodium acetate/acetic acid (NaOAc/HOAc; pH 4.5). Total RNA was isolated using acetate-saturated phenol/CHCl3 (pH 4.5; Thermo Fisher; cat. AM9722). RNA was resuspended in 10 mM NaOAc/HOAc (pH 4.5). Samples were split in two, one half (2 µg) was oxidized with 50 mM NaIO4 in 100 mM NaOAc/HOAc (pH 4.5) for 15 min and the other half (2 µg) was incubated in 50 mM NaCl in 100 mM NaOAc/HOAc (pH 4.5) for 15 min. Samples were quenched with glucose 100 mM for 5 min at room temperature, purified in G50 columns (Cytiva; cat. 28903408), and then precipitated with ethanol. tRNAs were deacylated in 50 mM Tris-HCl (pH 9.0) for 30 min at 37 °C. RNA was precipitated and then ligated to the 3′ adapter tRNA using T4 RNA ligase 2 (NEB; cat. M0351S). for 2 h at 37 °C. Reverse transcription was performed with the high processivity SuperScript IV synthesis kit (Thermo Fisher; cat. 18091050) at 60 °C for 15 minutes. To confirm that no stalling or premature termination occurred, full length tRNA sequences were validated by Sanger sequencing. Relative aminoacylation levels were calculated by qRT-PCR using tRNA specific primers (Supplementary Data 1).

## tRNA-Ribosome association assay

Cells were washed with ice-cold PBS supplemented with 100 µg/ml of CHX and lysed using NP40 lysis buffer. Ribosomes were immunoprecipitated with GFP-Trap, RFP-Trap, or NeonGreen-Trap Magnetic beads (Chromotek), as described above. RNA was isolated using saturated phenol/CHCl3 (pH 4.5) (Thermo Fisher; cat. AM9722). 500 ng of precipitated RNA was deacylated in 50 mM Tris-HCl (pH 9) for 30 minutes at 37 °C. The RNA was then precipitated and ligated to the 3′ adapter tRNA using T4 RNA ligase 2 (NEB; cat. M0351S) for 2 hours at 37 °C. Reverse transcription and sequence verification were performed under the same conditions as the tRNA aminoacylation assay. Primers and linker sequences are provided in Supplementary Data 1.

## Illumina sequencing

Ribo-Seq libraries were sequenced on an Illumina NextSeq 2000 (P2 flow cell, 100-cycle kit) at the DKFZ Sequencing Open Lab. Libraries (2 nM) were diluted to 650 pM by combining 7.8 µL library with 16.2 µL Illumina RSB. A 2% PhiX control spike-in was included by adding 1 µL of 1 nM PhiX. Sequencing was run in single-read mode for 110 cycles.

## Sequencing data analysis

For data preprocessing and alignment, the adapter sequences were trimmed using cutadapt (v3.4) and demultiplexed with barcode splitter from FASTX-toolkit (Cold Spring Harbor Laboratory, Hannon Lab, by Assaf Gordon). rRNA and tRNA sequences were filtered by alignment to indices of rRNA and tRNA sequences respectively, using BLAST-Like Alignment Tool (BLAT). A rRNA index was constructed from GENCODE v19 annotations, transcript types "rRNA", "Mt_rRNA" and "rRNA_pseudogene", supplemented with UCSC repeats of class "rRNA". The tRNA index was constructed from sequences obtained from GtRNAdb25 at June 2023. Unique molecular identifiers (UMIs) were extracted with umi_tools (v1.1.1) and the rRNA discarded with BLAST-Like Alignment Tool (BLAT, v36x2). The remaining reads were aligned to the GRCh37/hg19 human genome or to GRCm38/mm10 mouse genome by Spliced Transcripts Alignment to a Reference (STAR, v2.5.3a). PCR duplicates were removed for differential gene expression analysis using umi tools. The data was subjected to differential expression analysis with DESeq2 (v1.8.2) and transcripts sorted by fold change with a Benjamini-Hochberg adjusted $p$-value of $\leq 0.05$. Gene Ontology enrichment (GO term) analysis was conducted with clusterProfiler (v3.14.3) and R (v3.6.2).

Subsequence shift analysis involves comparing the frequencies of codon occupancy by ribosome-protected fragments (RPFs) between different samples. This comparison is done at the gene level, normalizing for gene expression discrepancies to ensure that observed differences in codon frequencies are not due to variations in gene expression levels. Specifically, RPFs (>26 nt) were assigned gene IDs and reading frames using GENCODE v19/BASIC. RPFs falling outside valid coding sequences (considering the 15-nucleotide 5′-overhang), those with ambiguous gene IDs or reading frames, were excluded. The remaining RPFs were utilized to tally the frequencies of all codons at different positions (12 and 15 nucleotides from the 5′-end) across all genes in the transcriptome. Gene-specific frequencies were determined by dividing the observed counts by the total counts for each gene. These normalized codon frequencies were then averaged across genes with a minimum count threshold in both condition and control samples (set at 100 for all figures unless stated otherwise). Shifts in codon frequencies were calculated based on these normalized and averaged values, expressed as (condition − control)/control, providing a relative measure of codon shifts compared to the control. To assess significant differences in codon occupancy at the amino acid level, subsequence shifts between control and condition samples across replicates were examined. This analysis utilized a linear mixed model (implemented via the R

package 'lme4') with fixed effects for the 20 amino acids and random effects for codons, yielding t-values and $p$-values. Multiple testing correction was performed using the Benjamini−Hochberg method ('p.adjust' function in R with method set to "fdr") to obtain adjusted $p$-values.

For RPF density analysis, codon-regions spanning 61 nucleotides around specified codons throughout the transcriptome (as annotated by GENmethod v19/BASIC) were identified using transcript coordinates, ensuring that codon-regions were entirely contained within exons. Overlapping transcript annotations were accounted for, retaining overlapping regions while collapsing those with identical genomic coordinates. Codon-regions unable to extend to the full 61 nucleotides (typically near transcript ends) were excluded from further analysis. The 5′ ends of ribosome-protected fragments (RPFs) were counted within each codon-region. For comparative analysis between two samples, only codon-regions with a minimum count of n in both samples were considered (set at 50 for most figures, unless specified otherwise; occasionally lowered to include at least 1000 windows for common codons). Normalized 5′ end RPF density for each sample and codon-region was computed by dividing the total counts within the region by the region's width, ensuring an average density of 1 within each region. These normalized densities underwent convolution using a rectangular window of width 3 and height 1/3. The average density across codon-regions was calculated, and the disparity in mean densities between samples yielded the density shift.

## Out-of-frame analysis

To evaluate the statistical significance of codon-specific shifts, we generated a background distribution by shifting ±1 nucleotide relative to the codon under analysis. To minimize the influence of true signals, specific values were excluded from this background distribution. For instance, under glucose starvation, which produces a signal at the GCA codon in the 15th position, we excluded NGC and CAN from the 11th and 13th positions, respectively, while including other nucleotide triplets observed at these positions. This refined background distribution was then used to determine the significance of the diricore signal at the original (0) position using Z-tests. Additionally, we verified that the background values approximated a normal distribution through visual inspection of Q-Q plots, as well as the Anderson−Darling and Shapiro−Wilk tests.

## Fluorescence microscopy and image analysis

Cells were plated on 8-well chambered coverslips (IBIDI) and left to attach overnight. Lysosomal proteolysis was investigated by live cell imaging with DQ BSA green or red (Thermo Fisher; cat. D12050 or D12051). In brief, cells were incubated with 0.1 mg/ml DQ BSA for 4−6 h, washed two times and chased for 3 h in fresh media to allow lysosomal accumulation of DQ BSA. 0.5 µg/ml Hoechst were added prior imaging. Imaging was performed in a humidified chamber at 37 °C and 5% CO2 with a Leica TCS SP5 confocal microscope using a ×40 or ×63, 1.40 oil objective. Fluorescence was quantified using the particle analyzer function of Fiji in randomly chosen fields of view across the entirety of each sample. Mean cellular fluorescence was determined by normalizing the integrated signal density of the respective fluorescent probe to cell number. To quantify lysosomal DQ BSA fluorescence dequenching, DQ BSA integrated density was normalized to the total cell area.

## RNA isolation and quantitative real-time PCR

Total RNA isolation was performed via phenol-chloroform extraction using TRI Reagent (Zymo Reseach; cat. R2050-1-200), followed by ice-cold isopropanol precipitation and centrifugation at 20,000 x g for 45 min at 4 °C. RNA pellet was washed twice with 75% ethanol, and resuspended in nuclease-free water. The concentration was determined at 280 nm with NanoDrop One C (Thermo Fisher). 600 ng of

total RNA was reversed transcribed using LunaScript RT Supermix (NEB; cat. M3010L) and quantitative real-time PCR was performed using Luna Universal qPCR Mix (NEB; cat. M3003X). Ct values were obtained with the Quantstudio 5RT qPCR System and analyzed using Quantstudio Design and Analysis Software 2.6.0. mRNA fold change of target genes was calculated by the ΔΔCt method. mRNA expression was normalized to GAPDH in human samples and β-Actin in mouse samples (Supplementary Data 1).

## Flow cytometry and flow sorting

Flow cytometry was performed on the BD Canto or Fortessa HTS FACS machines. FACS sorting was conducted on a BD Aria II and flow analyses were performed with FlowJo 10.6.1 software (LLC). For rapid separation of CD8$^+$ T cells and tumor cells from co-culture plates, CD8 T cells were transferred from the co-culture plates to a 10 μm cell strainer (PluriSelect) to retain any tumor cells detached from the plate[28]. Staining with 1:100 APC anti-mouse CD8, clone 53-6.7 (Thermo Fisher; cat. 17-0081-82) and 1:1000 DAPI (Thermo Fisher) was carried out for 1 h at 37 °C. Prior to sorting, events were gated sequentially as follows: (1) FSC-A vs SSC-A to exclude debris; (2) FSC-A vs FSC-H to select singlets; (3) DAPI– to exclude dead cells; and (4) APC vs GFP to distinguish CD8$^+$ T cells (APC$^+$) from cancer cells (GFP$^+$). Living cells were sorted with a 100 μm nozzle for GFP (cancer cells) and APC (T cells).

P14:Cas9 CD8 T cells with KO of SHMT1 or PHGDH were sorted based on the expression of Cas9-GFP and expression of the BFP reporter protein coded on the sgRNA transfer vector after transduction. Gates were set on FSC/SSC to select lymphocyte-sized singlets, DAPI– to exclude dead cells, and then on GFP$^+$ and BFP$^+$ quadrants using single-color controls. To quantify tumor-infiltrating lymphocytes, tumors were digested using the Tumor Dissociation Kit (Miltenyi Biotec; cat. 130-096-730). The cells were stained with anti-mouse CD8 antibodies (BD Biosciences; cat. 557668, 1:200) or anti-mouse Ki-67 (BioLegend; cat. 652424, 1:200) and analyzed using the Fortessa HTS FACS instrument. Analysis gates included FSC-A vs SSC-A to exclude debris, FSC-A vs FSC-H to select singlets, and live/dead exclusion via DAPI; CD8$^+$ or Ki-67$^+$ populations were then quantified against appropriate fluorescence-minus-one controls.

## CD8 + T cell isolation and culture

Primary naive CD8 + T cells were purified using the MojoSort Mouse CD8 T cell isolation kit (BioLegend; cat. 480007) and activated for 72 h on plates coated with 2 μg/ml anti-CD3 (BioXCell; cat. BE0001-1) and 2 μg/ml anti-CD28 (BioXCell; cat. BE0015-1) for 72 hours at 37 °C. T cells were kept in RPM1 1640 (Thermo Fisher; cat. 21875-034) supplemented with 10% FBS and 1% penicillin-streptomycin, 1 mM sodium pyruvate (Thermo Fisher; cat. 11360-070), 20 mM HEPES (Sigma; cat. H4034), 50 μM β-mercaptoethanol (Sigma; cat.M6250) and 10 ng/ml murine IL-2 (Biolegend; cat. 575404). For serine/glycine starvation experiments, medium was prepared fresh from RPMI 1640 powder without amino acids (US Biological; cat. R9010-01) and each component added individually according to the ATCC formulation. Human MART-1-specific (TCR clone #1D3) CD8 + T cells were donated by Dr. Chong Sun and kept in culture as described by Miao, B. et al.[63].

## T cell knock out generation

Platinum-E cells were transfected with pMSCV vectors (Addgene: 102796) encoding a BFP selection marker and sgRNAs targeting PHGDH or SHMT1. Transfection used 10 μg DNA and PEI Max (Polyscience; cat. 24765-1) for 16 h at 37 °C, 5% CO$_2$. Cells were then cultured 8 h in RPMI containing 3 mM sodium butyrate. Retroviral supernatants were harvested at 48 h and 72 h post-transfection and filtered (0.45 μm). Naïve CD8$^+$ T cells were isolated from P14 × CD4-Cre × Rosa-LSL-Cas9-GFP mice using MojoSort Mouse CD8 T Cell Isolation Kit (BioLegend; cat. 480007). Cells were activated on plates coated with 0.5 μg/mL anti-CD3 (BioXCell BE0001-1) and 2.5 μg/mL anti-CD28 (BioXCell; cat. BE0015-1) in RPMI supplemented with 1 ng/mL IL-2 (BioLegend; cat. 575404), 5 ng/mL IL-7 (PeproTech; cat. 217-17), 5 ng/mL IL-15 (PeproTech; cat. 210-15), and 55 μM β-mercaptoethanol. After 24 h, T cells were "spin-fected" with viral supernatant and Retronectin (Takara; cat. T100A) at 2,000 × g for 90 min at 30 °C, then incubated 24 h at 37 °C, 5% CO$_2$. Virus was removed by washing, and cells were expanded for 48 h in activation medium (1 μg/mL anti-CD3, 0.5 μg/mL anti-CD28, same cytokine concentrations).

## T cell killing and gene expression assays

CD8$^+$ OT-1 T cells were co-cultured with OVA-expressing mouse cancer lines (B16-OVA, TC1-OVA, E0771-OVA) in media containing increasing serine concentrations. After 24 h (37 °C, 5% CO$_2$), wells were washed with PBS, stained with crystal violet, and scanned on an Epson Dual Lens V850 Pro. Colony area (killing efficiency) was quantified using an ImageJ plugin[64]. For gene expression analysis, OT-1 cells and OVA$^+$ tumor cells were co-cultured at a 1:4 ratio for 24 h, then viable CD8$^+$ T cells were FACS-sorted. RNA was isolated and qPCR performed for serine-one-carbon metabolism genes. P14 T cells (PHGDH or SHMT1 KO) were co-cultured for 24 h with B16 cells expressing LCMV glycoprotein GP33 or with B16 cells pulsed with GP33 peptide (IBA 6-7016-901), then assessed by crystal violet staining as above. For human cells, MART-1–specific CD8$^+$ T cells were mixed with MART-1–expressing MDA-MB-231 cells at a 1:8 ratio. After 24 h, T cells were removed and tumor viability measured by CellTiter-Blue assay (Promega; cat. G8020).

## ELISA assays

Cytokines released by CD8$^+$ T cells were measured in culture supernatants using ELISA MAX™ Deluxe Mouse IFN-γ (BioLegend; cat. 430815) and Mouse Granzyme B ELISA (Thermo Fisher; cat. 88-8022-22) kits. Plates were read on a Thermo Fisher Multiskan FC at 450 nm with 570 nm reference subtraction. Concentrations were calculated in GraphPad Prism using a four-parameter logistic standard curve.

## Metabolomics

CD8 + T cells were isolated from the spleen of eight-week-old OT-1 female mice and activated as previously described[28]. For the co-culture assay, 1,000,000 E0771-OVA cells/replicate were plated in 15 cm dishes 48 hours before the co-culture to let them adhere. Pre-activated OT-1 CD8 + T cells were transferred on top of the E0771-OVA cells in a 1:5 ratio (CD8 + T cells: tumor cells). For T cell-only assays, 2,000,000 pre-activated CD8 + T cells/replicate were cultured in 6-well plates. For tumor-cell-only assays, 200,000 E0771-OVA cells/replicate were plated in 6-well plates 48 hours before the assay to let them adhere. Finally, 800,000 naïve CD8 + T cells/replicate were cultured in 24-well plates. All metabolic assays were conducted in full RPMI or RPMI without serine (Bioquote; cat. R9660). In both media, 12 C glucose was replaced by 13C6 labeled glucose (Cambridge Isotope Laboratories; cat. 110187-42-3). After six hours of culture in media with or without serine and in the presence of 13C6 glucose, CD8 + T cells and tumor cells were collected for further metabolic analysis. Collection, quenching, and metabolite extraction were performed as described by Elia, I. et al.[28]. Metabolites were measured using gas chromatography-mass spectrometry as described Rossi, M. et al.[65].

## Analysis of protein synthesis rates

Cells were incubated with 20 μM O-propargyl-puromycin (OPP; Jena Bioscience; cat. NU-931-05) for 1 hr at 37 °C, 5% CO$_2$[66]. After washing with PBS, cells were detached with trypsin-EDTA. In co-cultures, T cells and tumor cells were rapidly separated through a 10 μm cell strainer (PluriSelect)[28]. Cells were then fixed in 0.5 mL 1% (w/v) paraformaldehyde (Sigma-Aldrich; cat. 158127) in PBS for 15 min on ice (dark),

washed in PBS, and permeabilized for 5 min at room temperature in PBS containing 3% FBS and 0.1% saponin (Sigma-Aldrich; cat. SAE0073). OPP incorporation was detected by click chemistry: samples were incubated 30 min at room temperature (dark) with Click-iT Cell Reaction Buffer (Thermo Fisher; cat. C10269) and 5 µM Alexa Fluor 488- or 647-azide (Thermo Fisher; cat. A10266 or A10277). Excess dye was removed by two washes in PBS with 3% FBS and 0.1% saponin. Cells were resuspended in PBS and analyzed on a BD LSRFortessa (BD Biosciences). Median fluorescence intensities for each subpopulation were quantified using FlowJo v10.6.1.

### Transwell co-culture assay

OT-1 T cells were activated for 72 hours using plate-bound anti-CD3 (2 µg/mL) and anti-CD28 (2 µg/mL) in RPMI 1640 medium supplemented with 10% FBS, 1% penicillin-streptomycin, 1 mM sodium pyruvate, 20 mM HEPES, 50 µM β-mercaptoethanol, and 10 ng/mL murine IL-2. E0771 cells were seeded in the bottom chamber of a 24-well transwell plate (0.4 µm pore size, Corning; cat. 3450) and allowed to adhere overnight. Activated OT-1 T cells were then added to the upper chamber at a 2:1 T cell-to-tumor cell ratio. Co-cultures were maintained for 24 hours. After incubation, RNA was isolated from E0771 cells, and mRNA expression was assessed by qRT-PCR.

### Mice

6-8 weeks NSG, C57BL/6 J (Strain Code 027), OT-1, Rpl22 x CD4-Cre and P14 x CD4-Cre x Rosa-LSL-Cas9-GFP were either purchased from the Charles River Laboratory or bred in our animal facility[25,67–69]. Mice were housed under standard laboratory conditions. Experimental and control animals were co-housed under the same conditions to minimize environmental variables. A 12-hour light/12-hour dark cycle was maintained, with the ambient temperature regulated at $22 \pm 2$ °C and relative humidity set between 45% and 65%. Animals were provided with unrestricted access to food and water and were housed in groups of 3 to 5 per cage. All mice were kept in a pathogen-free environment and handled in accordance with protocols approved by the German Cancer Research Center and the supervisory authority (Regierungspräsidium Karlsruhe; G57-20), in compliance with the German Animal Protection Law and the European Directive 2010/63/EU on the protection of animals used for scientific purposes.

### Animal experiments

To assess checkpoint inhibition in the T cell–specific RiboTag model, CD4-Cre × Rpl22^HA/+ mice (8–10 weeks old) received $2 \times 10^6$ E0771–eGFP–RPL10a syngeneic breast cancer cells in the same site ($n \geq 4$ per group). Once tumors reached approximately 50–100 mm³, mice were treated with two intraperitoneal injections of anti–PD-1 antibody (2 µg per dose; BioXCell; cat. BP0273) or isotype control rat IgG (2 µg per dose; BioXCell; cat. BE0090), spaced three days apart. Tumor volume was measured every 2–4 days by caliper (volume = length × width²/2) until tumors reached a maximum volume of 1.5 cm³ or one dimension of 1.5 cm; tumors were then harvested, flash-frozen, and stored for DualRP library preparation.

For dietary intervention studies, female C57BL/6 J mice (7–8 weeks old; Charles River, strain code 027; $n \geq 6$ per group) were injected with $2 \times 10^6$ E0771–eGFP–RPL10a cells as above. Seven days post-injection, mice were switched to either a serine/glycine–free diet (TestDiet, cat. 5BQS) or matched control diet (TestDiet, cat. 5BQT). When tumors became palpable, mice received five intraperitoneal injections of anti–PD-1 or rat IgG (2 µg per dose) every two days. Tumor measurements were taken every 2–4 days.

Animals were euthanized by cervical dislocation when any of the following humane endpoints were met: tumor volume ≥ 1.5 cm³ or ≥1.5 cm in one dimension (tumor size was determined by caliper measurement), tumor ulceration, >20% weight loss relative to age-, sex-, strain-, and diet-matched controls, or signs of distress (apathy, reduced food/water intake, respiratory difficulty, or abnormal posture/movement). Animals were monitored every 2–4 days.

### Reporting summary

Further information on research design is available in the Nature Portfolio Reporting Summary linked to this article.

## Data availability

The sequence data from this study have been submitted to NCBI Bio-Project (http://www.ncbi.nlm.nih.gov/bioproject) under BioProject number PRJNA1024949. The metabolomics raw data have been deposited in Metabolomics Workbench under the Study ID ST003892 (https://doi.org/10.21228/M8MZ6D). All data are included in the Supplementary Information or available from the authors, as are unique reagents used in this Article. The raw numbers for charts and graphs are available in the Source Data file whenever possible. Source data are provided with this paper.

## Code availability

The code for differential ribosome codon reading (diricore) is available at https://github.com/A-X-Smitt/B250_diricore.

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

## Acknowledgements

We thank William Faller and Joana Silva for sharing reagents. We are grateful to Aurelio Teleman for comments on the manuscript. This work was funded in part by grants of the European Research Council "DualRP" (ERC StG No. 759579) and the German Research Foundation (DFG 504774163 and DFG 545215964) to F.L.-P. D.A.H., G.P., C.P. are supported by Cancer Transitional Research and EXchange Program (Cancer-TRAX) within the German-Israeli Helmholtz International Research School. European Research Council "DRILL" (ERC StG No. 101078722) to C.S. I.E. acknowledges funding from FWO (G065122N). A.K is supported by a fellowship of the Helmholtz International Graduate School. C.C.A.R. is supported by the DKFZ International Postdoc Program.

## Author contributions

F.L.-P. conceived the project, designed all the experiments, and wrote the manuscript. Methodology and data acquisition: D.A.H., R.D.P., A.K., A.H.B., G.P., E.S., C.C.A.R., S.D., E.R., C.P., N.S., V.C.V., H.E., S.D., S.C.-D.M., B.M., V.C.V., K.M.-D. and C.S. Manuscript revision: W.P., D.A-H., R.D.P. and F.L-P.

## Funding

## Competing interests

The authors declare no competing interests.
