## [Transparent Peer Review file · Nature Communications]

Dual Ribosome Profiling reveals metabolic limitations of cancer and stromal cells in the tumor microenvironment

Corresponding Author: Dr Fabricio Loayza-Puch

Version 0:

Reviewer comments:

Reviewer #1

(Remarks to the Author)

The manuscript by Aviles-Huerta et al develops and implements dual ribosome profiling to study RNA translation and tRNA usage in the setting of the tumor microenvironment, where amino acid flux may result in limitations in translational processes. As a reviewer, I found this paper to be clearly written, rigorous and compelling. It touches upon several highly exciting fields of research, namely the tumor microenvironment and RNA translation. While there are always experiments that one could suggest (when serving as a peer reviewer), in this case, I do not think any essential experiments are missing from this paper. The additional experiments that I can think of would likely be minimally impactful for the overall paper, would not be necessary to give credibility to the existing data shown here, and perhaps would be a distraction from the main storyline. This will be an exciting paper for the scientific field. I have only a few comments.

1. I found the title and abstract to emphasize the technology development a little too much. It is true that dual ribosome profiling is a development here. But I think the main storyline of the paper is fundamentally about biology, and it is not a tech dev paper that limits to verification of the technique.

2. I found the bioinformatics analyses of Ribo-seq data to be somewhat thin. That is, I think more could be done to characterize mRNAs that are differentially translated but not differentially transcribed. For example, most of this paper focuses on tRNAs that are profiled, but the authors could also do more with looking at mRNAs that are selectively translated when stimulated by the microenvironment. These mRNAs could be validated with proteomics and western blots to show upregulation of mRNA translation using dual RP that leads to protein abundances. I think this might strengthen some aspects of the paper. As is, the paper doesn't strongly enough demonstrate that dual RP provides useful mRNA translation changes that would not be observed by RNAseq. For example, if gene X is up in dual RP when tumor cells are co-cultured with fibroblasts, does this reflect a transcriptional or translational change for gene X?

3. The actual quality of some of the ribo-seq data is perhaps not especially strong. Sup Fig 2, panels C and D have strange abundances of reads <27nt in size, which we typically observe as contaminants resulting from low sample quality when we do riboseq. In addition, the % in frame reads is low in these experiments. This contrasts to Sup Fig 5 panels I and J where the % in frame reads around 28-29nt is much higher, and the fraction of reads in 28-30nt size is much more.

Reviewer #2

(Remarks to the Author)

The authors developed the Dual Ribosome Profiling allowing to investigate metabolic restrictions within the tumor microenvironment. Using this method, they identified different metabolic constraints in T cells as compared to cancer cells. Although the role of serine and glycine in T cell activation and proliferation has been already shown in vitro, they showed that T cells undergo serine and glycine restriction in vivo in a tumor model. This method is a useful tool and may be used and extended to other cell types and other diseases in the future.

The manuscript is well-written and provides key insights into T cell metabolic restriction in a cancer model. However, the parallel made between the in vitro model and the in vivo one including immune check point blockade is somehow confusing. Specific points to be addressed are listed below.

Major points:

- Is the limitation you describe for alanine and glycine specific to breast cancer cell lines (Figure 2 and Supp 3)? Or at least specific to TNBC cell lines?
 - It would be appreciated to have a better description of the syngeneic model used regarding the differences (if some are observed) between anti-PD1 treatment versus IgG control? (Figure 4)
 - o Are there any significant differences in terms of tumor growth at your endpoint?
 - o What about T cell proportion?
- The point here is that changes in the TME due to a-PD1 treatment could also alter T cell metabolism. If there is no difference it must be mentioned in the manuscript.
- The differences in labeled serine and glycine in T cells upon co-culture is indisputable. However, at this stage it's difficult to conclude about increased serine and glycine synthesis since none of the key enzyme expressions have been assessed (Figure 4). Here the hypothesis that labeled serine and glycine in T cells may come from increased uptake of labeled serine/glycine synthesized by the cancer cells and released in the media cannot be excluded. Analyzing phgdh, psat and shmt expression in T cells following co-culture has to be added here (instead of Figure 5) if the authors want to conclude on serine synthesis.
 - Increased serine and glycine synthesis upon antigen recognition is not convincing since it can be induced by other signaling molecules, including other cell-cell contacts or metabolites produced by cancer cells. To really point out the antigen recognition effect, it would be interesting to repeat this experiment using OVA-loaded antigen presenting cells instead of cancer cells. Also to confirm the required physical interaction the authors should repeat the co-culture using a transwell set up, activating T cells with anti-CD3 and anti-CD28.
 - The authors showed an increase in phgdh, psat and shmt expressions in vivo following PD1 blockade. Does this checkpoint inhibition in vitro exacerbate phgdh and shmt expression in the co-culture model? (Figure 5) The question here is whether PD1 signaling somehow inhibits serine synthesis (downregulating key enzyme expressions) in T cells during the co-culture.
 - Is the impact of PD1 blockade on serine synthesis in T cells is specific to PD1 signaling or CTL4 or other immune checkpoints have the same impact on serine metabolism?
 - The authors nicely showed in figure 5h that serine deprivation combined with aPD1 worsen tumor growth as compared to control diet. When combined with aPD1, does serine/glycine deprivation affect T cell infiltration, proliferation or does it affect mainly their functions?

Minor points:

- Line 170 – refer to Supp Figure 4f and not Supp Figure 4
- For better understanding of the experimental set-up, it would be more informative to switch the schema Supp Fig6a with the one in Fig4a since all the described experiments are the comparison between anti-PD1 and IgG control.
- The authors should clarify how they set up the endpoint for the syngeneic model. Usually, a volume is provided as a “limit size” for a tumor and not as a size in one dimension.

Reviewer #3

(Remarks to the Author)

The DualRP approach is conceptually useful and interesting, and the authors have done great quality control to validate it from a technical perspective. The authors use DualRP to find evidence for a somewhat surprising (see refs to literature in specific points) alanine/glycine tRNA charging reduction during glucose limitation for which rescue in co-culture depends on the interferon response and lysosome activity. However, the relevance of their IFN-mediated effect on tRNA charging is unclear. The IFN response is not specific to nutrient limitation, so the rescue of tRNA charging by IFN may be more of an “accident” than an evolved response to nutrient limitation. Also, how physiologically relevant is this co-culture IFN response? It seems unlikely that cells growing in tissues together in an organism have an IFN response – maybe an in vitro specific thing.

The authors also find an interesting in vivo phenotype that T cells, but not tumor cells, have signals consistent with ribosome stalling at serine and glycine codons during immunotherapy. This suggests that serine might represent an endogenous limitation for T cell function, which would be exciting, but the authors instead follow a thorough and convincing (though less surprising/therapeutically relevant) line of experiments demonstrating that serine is required for tumor-killing T cell function in vitro and in vivo.

Broadly, I have remaining doubts about the biological significance of the ribosome stalling signals they detail. They find stalling that is well correlated with changing tRNA levels upstream, which should be true. However, there is scant evidence on the downstream end that these limitations for tRNA charging and ribosome elongation have an impact on protein output. This is essential if we are to believe that they matter for protein synthesis. If ribosome stalling acts as a symptom of other metabolic problems (for example, as a clue that serine is functionally limiting for nucleotide synthesis, not protein synthesis), I certainly find that very interesting – however the authors maintain throughout, including in the discussion, that their stalling phenotypes reflect a true problem for protein output.

My initial comments on this manuscript were unaddressed so many points below about the validity of the authors' statements remain similar.

Specific points:

- Why are the other stalling signals of equal strength (in positive or negative direction) not discussed in Fig 2/4? Do the authors think these are not real signals, and if so, where do they come from? The statistical assessment of the ribosome stalling signal is not well described in legend and methods (why was this particular model chosen, how the model choice was evaluated, how are replicates treated within the model) and the signal to noise is not very convincing. I am particularly

confused by the traces of ribosome density in the window around the A-site (for example in 4B) – in most cases the peak at -15 appears to be offset by a large dip in signal directly after, raising the possibility that this is an “edge effect” which would be removed by averaging across a slightly larger window/smoothing.

- Regarding the mechanistic dissection of the co-culture effect on alanine/glycine tRNA charging - it seems like none of the microscopy DQ-BSA experiments are done in limiting glucose, which is an issue. The authors need to show that they pathways specifically modulate lysosome activity in the context of low glucose treatment to connect the dots in their mechanism. It is possible (and my guess is not improbable?) that glucose limitation activates lysosome activity even in the absence of IFN signaling.
- It is surprising to see limitations for alanine and glycine during glucose limitation, as these particular amino acids are usually produced in vast excess and net secreted by cells. This is worth discussing, particularly as it is also somewhat in conflict with literature. Rossiter et al 2021 found that low glucose was not sufficient to cause intracellular alanine limitation (it had to be combined with inhibition of mitochondrial pyruvate uptake). Olszewski et al 2022 figure 4b finds that blocking glucose uptake leads to no change to alanine levels and increases glycine levels. How do the authors think glycine becomes limiting if excess serine is available in the culture medium? It is improbable that autophagy is a more substantial source for glycine than excess extracellular serine.
- “DualRP elucidates a type I IFN-mediated pathway that enhances nutrient availability for protein synthesis.” < not accurate to say this without showing that overall protein synthesis rates are reduced in concert with alanine and glycine tRNA charging and then rescued in co-culture compared to mono-culture upon glucose limitation. Further, it is quite possible to rescue tRNA charging without rescuing protein synthesis if protein synthesis is still reduced indirectly by signaling pathways that respond to changing amino acid levels, like GCN2. The authors should discuss that it may not be true that ribosome stalling at alanine/glycine codons directly affects protein production – instead, these changes could alter signaling to regulate protein synthesis. Therefore the authors should throughout the manuscript avoid implying that these nutrients are directly/functionally limiting for protein output. They can instead say “...enhances amino acid availability for tRNA charging.”
- “This evidence strongly supports the notion that both amino acids are limiting factors for an efficient T cell cytotoxic response.” < The specificity of this phenotype to T cells is interesting, but this statement should be toned down in the absence of evidence that increasing serine/glycine levels (through diet, for example) in vivo, OR PHGDH overexpression in T cells, improves immunotherapy outcome/tumor killing. The authors can only conclude that S/G are required for an efficient T cell response (as they do at the end of this section), not limiting.
- The serine synthesis tracing data is not interpretable in co-culture as intra/extracellular serine pools are rapidly exchanged, so the labeled serine may have come from synthesis by the cancer cells, export by the cancer cells, and uptake by the T cells, rather than synthesis from glucose. So this: “...T cells exclusively activated serine and glycine synthesis from glucose” cannot be concluded. There is not a perfect way to address this but alternate explanations should be discussed. One possibility to support their conclusions, similar to the logic employed by Faubert et al Cell 2017 (but opposite effect), is to show upstream labeling pattern of metabolites in between glucose and serine to show serine substrate labeling also increases similarly.
- The authors refer to measurements of tRNA charging throughout. However, uncharged tRNA is also present in the ribosome, particularly during starvation, and as I understand it, would also be detected in their measurement. It would thus be most accurate to call it a tRNA-ribosome association measurement. This is not necessary but they could explore total tRNA levels in their input lysate before ribosome pulldown to assess whether these effects are driven by any tRNA abundance changes. This may be relevant as Banh et al 2020 found serine limitation led to total serine tRNA level differences but no strong charging phenotype, and Hsu et al 2023 found arginine limitation reduced arginine tRNA levels.
- PC1 has to explain a greater proportion of the variance than PC2. This needs to be addressed. It also looks like something is off because the points are similarly separated across PC1 and 2 but they account for vastly different amounts of the variance in the samples.
- Fig2e should be displayed as points (ideally an entire gene set) highlighted on a volcano plot to clarify how these differences compare to everything else changing.
- “DQ BSA fluorescence and ribosome association of aminoacylated Ala- and Gly-tRNAs were strongly decreased in LYSET-deficient cells upon co-culture with 186 MRC5 cells (Fig. 2f-h;...” – typo, should say Fig 3f-h.
- The authors find serine codons are equally affected by serine/glycine limitation. This seems to conflict with recent work by Bahn et al 2020 showing that only specific codons (TCC/TCT) are affected by serine limitation. Can the authors discuss this?

Version 1:

Reviewer comments:

Reviewer #1

(Remarks to the Author)

The revised manuscript has addressed my comments. I do not have further comments.

Reviewer #2

(Remarks to the Author)

Most of my comments have been addressed and the figures and discussion have been amended accordingly. However, some points still need to be detailed/addressed:

- In the transwell setup, have the OTI cells been activated with aCD3/aCD28 (revised Supp Fig 6I)?

Please add this in the figure legend and add the details of the transwell experiment in the material and method section.

- One of my comments was to describe the impact of both ser/gly deprivation and a-PD1 treatment on T cell functions. The

authors provided answers for T cell infiltration and proliferation only. Are there any limitations in their model that prevent the authors from analyzing T cell functions (cytokine and cytotoxic molecule production) when isolated from the tumor (revised Supp Fig 8f)?

Reviewer #3

(Remarks to the Author)

The authors have addressed my major concerns excepting the following points on which I still disagree. I would advocate for minor changes to address these remaining points before publication, which I otherwise support.

1. "Regarding the dip observed at position -12 in the ribosome density plots, this feature reflects a reduced ribosome density at the P site of the codon, which aligns with transient stalling of ribosomes at the A site (position -15)." I don't follow the logic/interpretation here and I don't think this should be labeled as a fact in the legend, particularly without indicating the P/E sites on the plot. I think the authors are saying that because there is more ribosome density where that codon is exactly in the A-site, there is necessarily less where that codon is in the P-site. Why would this not apply to the E-site also? Also, their traces don't appear to be this precise – for the Ala codons, the dip doesn't occur, and for Ser codons, the dip occurs at position 0. This P-site dip is also not observed in other papers that show significant A-site specific ribosome stalling (see Ishimura et al 2016). I am honestly not convinced regarding the Diricore analysis that this peak is not in the same realm as the technical noise. However, the authors follow up data, particularly the ribosome-tRNA association, is interesting and valid and supports their ideas enough to let the Diricore stand – though I would consider acknowledging the limitations of this method with respect to signal to noise to avoid misleading others. There has been great debate and optimization in the ribosome profiling field about how to accurately capture ribosome positions with high resolution at codon resolution, and what artifacts might be inevitable due to library or harvesting methods, so I think the authors would really benefit from care here.
2. "Combining both treatments resulted in a robust increase in lysosomal activity in mono-cultures, as shown in Extended Data Fig. 4e." – I think they mean Ex Data Fig 4g, and I agree this data addresses my concern.
3. "TNBC cells lacking PHGDH show a heightened dependency on extracellular serine. In these cells, serine is not only taken up from the environment but is also synthesized from glycine through the action of serine hydroxymethyltransferase (SHMT)3 . This conversion process becomes crucial when de novo synthesis from glucose is compromised." This is an interesting point and would be easy to test by showing that PHGDH overexpression rescues Ala and Gly stalling upon low glucose. Not within the scope of the current study, I agree.
"Consistent with our Diricore analysis, we observed a significant decrease in the intracellular concentrations of alanine and glycine." I remain skeptical that a ~2-fold decrease in intracellular alanine and glycine pools is enough to limit protein synthesis (as the authors observe with proline). Do the authors propose that alanine and glycine tRNA charging has a significantly higher K_M than proline tRNA charging? This data should definitely be included in the manuscript and discussed for transparency.
4. I appreciate all of the changes the authors made to phrasing to not oversell the point that these amino acids are limiting protein production, that serine is limiting for the T cell response, and that they are measuring a tRNA-ribosome association and not a charged tRNA-ribosome association.

Version 2:

Reviewer comments:

Reviewer #2

(Remarks to the Author)

The authors have addressed all my comments/concerns. I do not have further comments.

Reviewer #3

(Remarks to the Author)

The authors have addressed all of my comments, I thank them for their thoughtful responses.

Response to Reviewers' Comments

We thank the Reviewers for their critical reading of the manuscript and are pleased they found the study compelling and interesting. In response to the helpful suggestions, we now provide substantial new data strengthening our conclusions. Below are detailed responses for each comment.

Reviewer #1 (Remarks to the Author):

The manuscript by Aviles-Huerta et al develops and implements dual ribosome profiling to study RNA translation and tRNA usage in the setting of the tumor microenvironment, where amino acid flux may result in limitations in translational processes. As a reviewer, I found this paper to be clearly written, rigorous and compelling. It touches upon several highly exciting fields of research, namely the tumor microenvironment and RNA translation. While there are always experiments that one could suggest (when serving as a peer reviewer), in this case, I do not think any essential experiments are missing from this paper. The additional experiments that I can think of would likely be minimally impactful for the overall paper, would not be necessary to give credibility to the existing data shown here, and perhaps would be a distraction from the main storyline. This will be an exciting paper for the scientific field. I have only a few comments.

1. I found the title and abstract to emphasize the technology development a little too much. It is true that dual ribosome profiling is a development here. But I think the main storyline of the paper is fundamentally about biology, and it is not a tech dev paper that limits to verification of the technique.

Response:

Thank you for this comment. We appreciate your perspective and agree that the primary focus of the manuscript is the biological insights gained from using Dual Ribosome Profiling (DualRP), rather than merely the development of the technology itself. The study uncovers critical metabolic vulnerabilities in the tumor microenvironment (TME), particularly in the T cell compartment, and highlights the role of serine and glycine in the efficacy of immune checkpoint blockade therapy. These findings have important implications for the design of combinatorial therapies targeting metabolic constraints in the TME.

To address your concern, we revised the title and abstract to better emphasize the biological discoveries enabled by DualRP, while still acknowledging its utility as a tool. We adjusted the title to: "Dual Ribosome Profiling reveals metabolic limitations of cancer and stromal cells in the tumor microenvironment." This shifts the focus to the biological storyline while retaining recognition of the method as a key enabler. In the abstract, we similarly adjusted the phrasing to highlight the biological findings upfront. This revision keeps the focus on the biological story while maintaining a balanced acknowledgment of the methodological innovation.

2. I found the bioinformatics analyses of Ribo-seq data to be somewhat thin. That is, I think more could be done to characterize mRNAs that are differentially translated but not differentially transcribed. For example, most of this paper focuses on tRNAs that are profiled, but the authors could also do more with looking at mRNAs that are selectively translated when stimulated by the microenvironment. These mRNAs could be validated with proteomics and western blots to show upregulation of mRNA translation using dual RP that leads to protein abundances. I think this might strengthen some aspects of the paper. As is, the paper doesn't strongly enough demonstrate that dual RP provides useful mRNA translation changes that would not be observed by RNAseq. For example, if gene X is up in dual RP when tumor cells

are co-cultured with fibroblasts, does this reflect a transcriptional or translational change for gene X?

Response:

Thank you for your constructive feedback. While we agree that such analyses would add significant depth to understanding translational regulation within the tumor microenvironment (TME), this direction is outside the current scope of the manuscript, which focuses primarily on amino acid restrictions and their functional consequences as revealed by Dual Ribosome Profiling (DualRP). However, we appreciate the opportunity to address this point and clarify the current focus of our study.

Our primary goal was to validate DualRP as a robust tool to detect cell-type-specific metabolic limitations in the TME, particularly in response to immune checkpoint blockade. The primary emphasis was on using ribosome stalling as a proxy to identify amino acid limitations, which are tightly linked to translational regulation of the T cell compartment. Nevertheless, we acknowledge the broader potential of DualRP to uncover translational changes beyond tRNA profiling, including the selective translation of mRNAs.

To address your point, we have added the following considerations to the Discussion section of the manuscript: “While our current analyses focus on amino acid limitations, DualRP inherently captures ribosome footprints across mRNAs, enabling identification of translationally regulated genes. Future studies could leverage these capabilities to systematically identify mRNAs that exhibit differential ribosome occupancy independent of transcriptional changes. For example, ribosome occupancy could be correlated with protein expression using proteomics or validated via western blot analyses.”

3. The actual quality of some of the ribo-seq data is perhaps not especially strong. Sup Fig 2, panels C and D have strange abundances of reads <27nt in size, which we typically observe as contaminants resulting from low sample quality when we do riboseq. In addition, the % in frame reads is low in these experiments. This contrasts to Sup Fig 5 panels I and J where the % in frame reads around 28-29nt is much higher, and the fraction of reads in 28-30nt size is much more.

Response:

We agree that ensuring high-quality ribosome profiling (ribo-seq) data is crucial for the accurate interpretation of translational and ribosome stalling analyses. Regarding the concerns raised in Supplementary Figure 2, panels c and d, we acknowledge the atypical distribution of reads <27 nt and the lower percentage of in-frame reads observed in these experiments. These discrepancies are likely attributable to variations in library preparation, such as RNase I treatment and size selection.

To ensure robust and reproducible data, our Diricore analysis includes only ribosome-protected fragments (RPFs) that align to frame 0 and are >27 nt, effectively discarding any potential contaminants our out-of-frame reads. We have now added a more detailed explanation of these parameters in the Methods section of the revised manuscript to address this concern transparently.

Reviewer #2 (Remarks to the Author):

The authors developed the Dual Ribosome Profiling allowing to investigate metabolic restrictions within the tumor microenvironment. Using this method, they identified different metabolic constraints in T cells as compared to cancer cells. Although the role of serine and glycine in T cell activation and proliferation has been already shown in vitro, they showed that T cells undergo serine and glycine restriction in vivo in a tumor model. This method is a useful tool and may be used and extended to other cell types and other diseases in the future.

The manuscript is well-written and provides key insights into T cell metabolic restriction in a cancer model. However, the parallel made between the in vitro model and the in vivo one including immune check point blockade is somehow confusing. Specific points to be addressed are listed below.

Major points:

- Is the limitation you describe for alanine and glycine specific to breast cancer cell lines (Figure 2 and Supp 3)? Or at least specific to TNBC cell lines?

Response:

In this study, we focused on TNBC models to demonstrate the utility and robustness of the DualRP technique under metabolically challenging conditions. While the alanine and glycine limitations were observed within these TNBC models (as shown in Figure 2 and Supp. 3), we acknowledge that it remains unclear whether these findings are unique to TNBC or generalizable to other breast cancer subtypes and tumor types.

We agree with the reviewer that expanding this analysis to additional cancer cell lines, including non-TNBC breast cancer models and other tumor types, would be highly informative. At present, our data do not allow us to conclude whether the observed phenotype can be generalized across all TNBC cell lines or other cancer types. In future studies, we aim to apply DualRP to a broader range of cancer models, such as lung adenocarcinoma and melanoma, to determine whether alanine and glycine limitations—and their alleviation through stromal interactions—are conserved across different tumor types.

This expanded approach will provide valuable insights into the context-specific nature of these metabolic restrictions and their broader implications for tumor-immune-stromal interactions. We thank the reviewer for highlighting this important point, which opens up exciting avenues for future investigation.

- It would be appreciated to have a better description of the syngeneic model used regarding the differences (if some are observed) between anti-PD1 treatment versus IgG control? (Figure 4)

o Are there any significant differences in terms of tumor growth at your endpoint?

o What about T cell proportion?

The point here is that changes in the TME due to a-PD1 treatment could also alter T cell metabolism. If there is no difference it must be mentioned in the manuscript.

Response:

We thank the reviewer for highlighting these points. Below, we provide detailed responses and outline how we have addressed the feedback to improve the manuscript:

In our syngeneic model using E0771 cells, we observed a significant reduction in tumor burden at the experimental endpoint in the anti-PD1 treatment group compared to the IgG control group in mice fed both full and (-) SG diets. This result aligns with the well-documented effects of immune checkpoint blockade in enhancing T cell-mediated anti-tumor responses. We have revised the manuscript to include a more detailed description of tumor growth dynamics, accompanied by statistical analyses to illustrate the differences between the groups (**Figure 5f and its corresponding figure legend**). These additions enhance clarity and strengthen the narrative regarding the impact of anti-PD1 treatment.

To quantify the proportion of T cells in the tumor microenvironment (TME), we conducted flow cytometry analysis. Our data indicate that anti-PD1 treatment increased the proportion of T cells compared to the IgG control, as assessed by flow cytometric analysis of CD8+ T cells, which are critical mediators of the cytotoxic response. Notably, the (-) SG diet reduced the proportion of infiltrating T cells in response to anti-PD1 treatment, suggesting that systemic

serine and glycine are required for an effective immunotherapeutic response. These findings have been incorporated into **Supplementary Figure 8f** in the revised manuscript.

To ensure completeness, we expanded the Methods section to include a more detailed description of the syngeneic model, covering the experimental design, tumor inoculation protocols, and timeline for anti-PD1/IgG administration. Additionally, we enhanced the Results section by discussing the observed differences in tumor growth and T cell proportions in the context of anti-PD1 treatment.

- The differences in labeled serine and glycine in T cells upon co-culture is indisputable. However, at this stage it's difficult to conclude about increased serine and glycine synthesis since none of the key enzyme expressions have been assessed (Figure 4). Here the hypothesis that labeled serine and glycine in T cells may come from increased uptake of labeled serine/glycine synthesized by the cancer cells and released in the media cannot be excluded.

Analyzing phgdh, psat and shmt expression in T cells following co-culture has to be added here (instead of Figure 5) if the authors want to conclude on serine synthesis.

Response:

We acknowledge that our current data do not explicitly distinguish between the increased synthesis of serine and glycine within T cells and their uptake from extracellular sources, including those synthesized and secreted by cancer cells. While our findings demonstrate that T cells activate serine and glycine biosynthetic pathways upon antigen recognition, we cannot entirely exclude the possibility of increased uptake of labeled serine and glycine from cancer cells at this stage.

To address this limitation, we analyzed the expression of PHGDH, PSAT1, and SHMT1/2 in T cells following co-culture with cancer cells. These enzymes are critical components of the serine biosynthesis pathway and one-carbon metabolism. We observed increased expression of all these genes in OT-I T cells after co-culture with cancer cell lines expressing OVA. This expression analysis complements our existing data on labeled serine and glycine incorporation, providing a mechanistic basis to support our conclusions.

To improve clarity and cohesion, we have relocated the analysis of PHGDH, PSAT1, and SHMT1/2 expression from Figure 5 to **Figures 4g–h** and **Supplementary Figures 7a–b** in the revised manuscript. This reorganization integrates these findings more effectively within the context of the co-culture experiments, enhancing the logical flow and coherence of the data presentation.

- Increased serine and glycine synthesis upon antigen recognition is not convincing since it can be induced by other signaling molecules, including other cell-cell contacts or metabolites produced by cancer cells. To really point out the antigen recognition effect, it would be interesting to repeat this experiment using OVA-loaded antigen presenting cells instead of cancer cells. Also to confirm the required physical interaction the authors should repeat the co-culture using a transwell set up, activating T cells with anti-CD3 and anti-CD28.

Response:

We thank the reviewer for their suggestion regarding the importance of distinguishing the specific contributions of antigen recognition from other cell-cell interactions or metabolites in our experimental system. We fully agree that incorporating additional controls and alternative setups will enhance the mechanistic clarity of our findings.

To directly address the reviewer's concern about antigen specificity, we performed co-culture experiments involving CD8⁺ OT-I T cells and SIINFEKL peptide-loaded DC 2.4 antigen-presenting cells (APCs) for 24 hours. Following co-culture, we separated the two cell

populations as described in Supplementary Fig. 6c and analyzed the expression of genes involved in the serine biosynthesis pathway (PHGDH, PSAT1, and PSPH), as well as those facilitating serine entry into one-carbon metabolism (SHMT1 and SHMT2). Our results show a significant upregulation of all these genes in OT-I T cells co-cultured with SIINFEKL-pulsed APCs compared to non-pulsed APCs (**Supplementary Fig. 6k**).

Furthermore, to confirm the required physical interaction, we used a transwell setup to co-culture OT-I T cells with E0771-OVA cancer cells, thereby preventing direct physical contact while allowing the diffusion of soluble factors between compartments. In this setup, we observed no upregulation of PHGDH, PSAT1, PSPH, SHMT1, or SHMT2 transcripts in OT-I T cells, indicating that physical interaction is necessary to drive serine and glycine metabolism. These results have been incorporated into **Supplementary Fig. 6l** of the revised manuscript.

- The authors showed an increase in phgdh, psat and shmt expressions in vivo following PD1 blockade. Does this checkpoint inhibition in vitro exacerbate phgdh and shmt expression in the co-culture model? (Figure 5)

The question here is whether PD1 signaling somehow inhibits serine synthesis (downregulating key enzyme expressions) in T cells during the co-culture.

Response:

To directly address this, we conducted additional experiments co-culturing OT-I CD8⁺ T cells with E0771-OVA cancer cells, with and without PD-1 blockade using α PD-1 antibodies. After 24 hours of co-culture, we isolated both cell populations and measured the transcript levels of PHGDH, PSAT, PSPH, and SHMT1 in OT-I T cells via qPCR. Our in vitro results indicate that α PD-1 treatment does not significantly affect the expression of genes involved in serine synthesis or the entry into one-carbon metabolism pathways (**Figure R1**). These findings suggest that the limitation of serine and glycine within the tumor microenvironment is likely due to the activation and expansion of T cells upon α PD-1 treatment (**Supplementary Figure 8f**).

Figure R1. Quantification of the indicated genes using qRT-PCR in co-cultures of OT-I CD8⁺ T cells and E0771-OVA cells treated with IgG (2 μ g/ μ l) or α PD-1 (2 μ g/ μ l). After 24 hours of co-culture, cells were separated by size (see Supplementary Fig. 6c). Data are presented as mean \pm SD. NS, not significant.

- Is the impact of PD1 blockade on serine synthesis in T cells is specific to PD1 signaling or CTL4 or other immune check points have the same impact on serine metabolism?

Response:

While our study primarily examines the effects of PD1 signaling on T cell metabolism, we acknowledge that CTLA-4 and other immune checkpoints may similarly impact serine metabolism due to their overlapping roles in modulating T cell activation and metabolic reprogramming.

Our findings demonstrate that serine and glycine are required for an efficient immunotherapy response, highlighting a metabolic adaptation to immune checkpoint inhibition. These results were validated both in vivo (using DualRP) and in co-culture systems. However, this study did not directly investigate the specific role of CTLA-4 or other immune checkpoints in regulating serine metabolism.

In the revised manuscript, we have included a discussion on whether the observed effects on serine biosynthesis are unique to PD1 signaling or if similar metabolic adaptations may occur with CTLA-4 blockade or other checkpoint inhibitors. We reference existing literature that outlines overlapping and distinct downstream signaling events triggered by PD1 and CTLA-4 inhibition and their respective impacts on T cell metabolism. Furthermore, we propose potential experimental approaches for future research, such as applying DualRP to T cells from tumors treated with CTLA-4 inhibitors or combination therapies, to clarify the specificity of serine metabolism changes across checkpoint pathways.

These additions provide a broader context for interpreting our findings and offer direction for future studies on the interplay between immune checkpoint blockade therapies and the metabolic requirements of T cells.

- The authors nicely showed in figure 5h that serine deprivation combined with aPD1 worsen tumor growth as compared to control diet. When combined with aPD1, does serine/glycine deprivation affect T cell infiltration, proliferation or does it affect mainly their functions?

Response:

To investigate the effects of serine and glycine deprivation on T cell proliferation and function in vivo, we performed additional flow cytometry experiments to quantify the frequency of tumor-infiltrating CD8+ T cells in mice treated with α PD1 under control and serine/glycine-deprived diets. Our findings demonstrated that anti-PD1 treatment increased the proportion of T cells within the tumor microenvironment (TME) compared to the IgG control, indicating an expansion of the T cell population. However, serine/glycine deprivation markedly reduced the proportion of infiltrating T cells in response to anti-PD1 treatment. This reduction was associated with impaired T cell proliferation, as indicated by decreased Ki67 expression in CD8+ T cells. These results suggest that serine and glycine deprivation negatively affect T cell proliferation in vivo during anti-PD1 treatment. These data are now included in **Supplementary Fig. 8f** of the revised manuscript.

Minor points:

- Line 170 – refer to Supp Figure 4f and not Supp Figure 4

Thank you for pointing this out. We have corrected the reference in Line 170 to specify Supplementary Figure 4f.

- For better understanding of the experimental set-up, it would be more informative to switch the schema Supp Fig6a with the one in Fig4a since all the described experiments are the comparison between anti-PD1 and IgG control.

We appreciate this suggestion and agree that it will improve clarity. We have switched the schemas as recommended to better align with the experimental comparisons described.

- The authors should clarify how they set up the endpoint for the syngeneic model. Usually, a volume is provided as a “limit size” for a tumor and not as a size in one dimension.

Thank you for highlighting this. We have now clarified the endpoint criteria in the Methods section, specifying that tumor volume limits were used to define the endpoint and providing the exact volume threshold as a guideline.

Reviewer #3 (Remarks to the Author):

The DualRP approach is conceptually useful and interesting, and the authors have done great quality control to validate it from a technical perspective. The authors use DualRP to find evidence for a somewhat surprising (see refs to literature in specific points) alanine/glycine tRNA charging reduction during glucose limitation for which rescue in co-culture depends on the interferon response and lysosome activity. However, the relevance of their IFN-mediated effect on tRNA charging is unclear. The IFN response is not specific to nutrient limitation, so the rescue of tRNA charging by IFN may be more of an “accident” than an evolved response to nutrient limitation. Also, how physiologically relevant is this co-culture IFN response? It seems unlikely that cells growing in tissues together in an organism have an IFN response – maybe an in vitro specific thing.

The authors also find an interesting in vivo phenotype that T cells, but not tumor cells, have signals consistent with ribosome stalling at serine and glycine codons during immunotherapy. This suggests that serine might represent an endogenous limitation for T cell function, which would be exciting, but the authors instead follow a thorough and convincing (though less surprising/therapeutically relevant) line of experiments demonstrating that serine is required for tumor-killing T cell function in vitro and in vivo.

Broadly, I have remaining doubts about the biological significance of the ribosome stalling signals they detail. They find stalling that is well correlated with changing tRNA levels upstream, which should be true. However, there is scant evidence on the downstream end that these limitations for tRNA charging and ribosome elongation have an impact on protein output. This is essential if we are to believe that they matter for protein synthesis. If ribosome stalling acts as a symptom of other metabolic problems (for example, as a clue that serine is functionally limiting for nucleotide synthesis, not protein synthesis), I certainly find that very interesting – however the authors maintain throughout, including in the discussion, that their stalling phenotypes reflect a true problem for protein output. My initial comments on this manuscript were unaddressed so many points below about the validity of the authors’ statements remain similar.

Specific points:

- Why are the other stalling signals of equal strength (in positive or negative direction) not discussed in Fig 2/4? Do the authors think these are not real signals, and if so, where do they come from? The statistical assessment of the ribosome stalling signal is not well described in legend and methods (why was this particular model chosen, how the model choice was evaluated, how are replicates treated within the model) and the signal to noise is not very convincing. I am particularly confused by the traces of ribosome density in the window around the A-site (for example in 4B) – in most cases the peak at -15 appears to be offset by a large dip in signal directly after, raising the possibility that this is an “edge effect” which would be removed by averaging across a slightly larger window/smoothing.*

Response:

We thank the reviewer for these comments. To enhance the statistical assessment and robustness of our diricore signal, we have undertaken the following actions:

Discussion of additional stalling signals

Regarding the other stalling signals of similar strength in Figures 2 and 4, we agree that a deeper exploration of these signals could provide valuable insights. In our analysis, we primarily focused on the serine and glycine codons as these emerged as significantly enriched and relevant to the central findings of our study. However, we acknowledge that other stalling events may represent biologically meaningful phenomena. To clarify this, we revised the manuscript to include a discussion of these additional signals, their potential biological origins, and our rationale for focusing on the highlighted signals.

Detailed description of the statistical analysis:

We have added a detailed description in the Methods section outlining our out-of-frame analysis to evaluate the statistical significance of codon-specific shifts. Briefly, we generated a background distribution by shifting ± 1 nucleotide relative to the codon under analysis. To minimize the influence of true signals, specific values were excluded from this background distribution. For instance, under glucose starvation, which produces a signal at the GCA codon in the 15th position, we excluded NGC and CAN from the 11th and 13th positions, respectively, while including other nucleotide triplets observed at these positions. This refined background distribution was then used to determine the significance of the diricore signal at the original (0) position using Z-tests. Additionally, we verified that the background values approximated a normal distribution through visual inspection of Q-Q plots, as well as the Anderson–Darling and Shapiro–Wilk tests. This model allows to compare the shifts of codons within a single replicate. Biological replicates are provided for the main findings in the manuscript.

We evaluated the model using ribosome profiling datasets of cancer cells treated with harringtonine and asparaginase, which generate signals at methionine and asparagine codons, respectively¹.

Clarification of signal patterns in the ribosome density plots:

Regarding the dip observed at position -12 in the ribosome density plots, this feature reflects a reduced ribosome density at the P site of the codon, which aligns with transient stalling of ribosomes at the A site (position -15). We have clarified this in the relevant figure legends for all diricore plots to provide clearer interpretation.

Noise reduction in extended Data Figure 3:

The noise in the subsequence plots of Fig 2 and Extended Data Figure 3 primarily stemmed from the lower read counts in the glucose-starved monoculture datasets. To address this, we re-sequenced the 10% glucose monoculture libraries and integrated these new reads with our existing data, followed by re-analysis with the diricore method. This approach significantly reduced noise and non-specific signals. Further out-of-frame analysis confirmed the statistical significance of the signals specifically at Alanine and Glycine codons. The new diricore analysis is incorporated in **Figure 2** and **Extended Data Figure 3**.

• Regarding the mechanistic dissection of the co-culture effect on alanine/glycine tRNA charging - it seems like none of the microscopy DQ-BSA experiments are done in limiting glucose, which is an issue. The authors need to show that they pathways specifically modulate lysosome activity in the context of low glucose treatment to connect the dots in their mechanism. It is possible (and my guess is not improbable?) that glucose limitation activates lysosome activity even in the absence of IFN signaling.

Response:

To determine the effect of limiting glucose concentrations on cancer cells lysosomal catabolism and whether this effect is STAT1- or LYSET-dependent, we conducted additional DQ-BSA degradation measurements. First, we performed DQ-BSA fluorescence assays in mono-cultures grown under low-glucose conditions or treated with IFN- β alone. We observed that low-glucose conditions in SUM-159PT cells did not increase lysosomal activity, whereas IFN- β treatment significantly enhanced it. Combining both treatments resulted in a robust increase in lysosomal activity in mono-cultures, as shown in **Extended Data Fig. 4e**.

Next, we evaluated the impact of STAT1 and LYSET knockouts on lysosomal activity in both mono- and co-cultures under full-medium and low-glucose conditions. Neither STAT1 nor LYSET knockouts increased the DQ-BSA signal in SUM-159PT cells grown as mono-cultures. However, both knockouts significantly reduced lysosomal activity when SUM-159PT cells were co-cultured with MRC5 fibroblasts in full medium (**Fig. 3c-d**; **Fig. 3f-g**). This reduction

was consistent even under limiting glucose concentrations (**Extended Data Fig. 4h and 4k**). These findings support the conclusion that co-culturing breast cancer cells with fibroblasts enhances lysosomal catabolism in an IFN-I-dependent manner.

• It is surprising to see limitations for alanine and glycine during glucose limitation, as these particular amino acids are usually produced in vast excess and net secreted by cells. This is worth discussing, particularly as it is also somewhat in conflict with literature. Rossiter et al 2021 found that low glucose was not sufficient to cause intracellular alanine limitation (it had to be combined with inhibition of mitochondrial pyruvate uptake). Olszewski et al 2022 figure 4b finds that blocking glucose uptake leads to no change to alanine levels and increases glycine levels. How do the authors think glycine becomes limiting if excess serine is available in the culture medium? It is improbable that autophagy is a more substantial source for glycine than excess extracellular serine.

Response:

We thank the reviewer for this comment. Our study uses the triple-negative breast cancer (TNBC) cell lines SUM-159PT and MDA-MB-231 as a model, both of which lack the expression of phosphoglycerate dehydrogenase (PHGDH)². TNBC cells lacking PHGDH show a heightened dependency on extracellular serine. In these cells, serine is not only taken up from the environment but is also synthesized from glycine through the action of serine hydroxymethyltransferase (SHMT)³. This conversion process becomes crucial when de novo synthesis from glucose is compromised. The reliance on serine derived from glycine underscores the metabolic flexibility of TNBC cells, allowing them to adapt to varying nutrient availabilities while maintaining essential biosynthetic pathways.

Regarding the conflicting findings from Rossiter et al. (2021) and Olszewski et al. (2022), it is important to note that while their studies indicate that low glucose alone does not cause alanine limitation, our observations suggest that the interplay between glucose availability and other metabolic pathways—such as those involving PHGDH—may lead to distinct outcomes in different cellular contexts. For example, while some studies report the accumulation of glycine when glucose uptake is blocked, this may reflect a compensatory mechanism in certain cell types rather than a universal response across all TNBC cells. We address these points in the Results section of the revised version of our manuscript.

Furthermore, to experimentally test this, we measured intracellular amino acid levels in SUM-159PT cells under full medium conditions and after glucose starvation. Consistent with our Diricore analysis, we observed a significant decrease in the intracellular concentrations of alanine and glycine. Additionally, we noted a significant decrease in proline levels. However, our Diricore analysis did not show any proline limitation following glucose starvation, suggesting that although there is a decrease in amino acid levels, it is not limiting for protein synthesis (Figure R2).

Figure R2. Heatmap representing changes in amino acid levels, measured by LC-MS, in SUM-159PT cells grown in full medium (100% glucose, 25 mM) or glucose starvation conditions (10% glucose, 2.5 mM) for 48 hours. The data show the average of three biological replicates. For each amino acid, the values in 100% glucose medium are presented as log₂ (fold change) relative to the values in 10% glucose medium.

• “DualRP elucidates a type I IFN-mediated pathway that enhances nutrient availability for protein synthesis.” < not accurate to say this without showing that overall protein synthesis rates are reduced in concert with alanine and glycine tRNA charging and then rescued in co-culture compared to mono-culture upon glucose limitation. Further, it is quite possible to rescue tRNA charging without rescuing protein synthesis if protein synthesis is still reduced indirectly by signaling pathways that respond to changing amino acid levels, like GCN2. The authors should discuss that it may not be true that ribosome stalling at alanine/glycine codons directly affects protein production – instead, these changes could alter signaling to regulate protein synthesis. Therefore the authors should throughout the manuscript avoid implying that these nutrients are directly/functionally limiting for protein output. They can instead say “...enhances amino acid availability for tRNA charging.”

Response:

We appreciate the reviewer’s comment. We agree that our data does not unequivocally demonstrate that alanine and glycine tRNA charging directly correlate with overall protein synthesis rates or that the rescue of tRNA charging necessarily implies a direct rescue of protein output under glucose-limiting conditions. In light of this, we have revised the manuscript to address this concern. Specifically, throughout the manuscript, we have reframed our conclusions to avoid directly linking nutrient availability to protein synthesis. For instance, we have replaced “enhances nutrient availability for protein synthesis” with “enhances amino acid availability for tRNA charging.” Furthermore, we have included a discussion exploring the possibility that ribosome stalling may not directly impact protein synthesis.

• “This evidence strongly supports the notion that both amino acids are limiting factors for an efficient T cell cytotoxic response.” < The specificity of this phenotype to T cells is interesting, but this statement should be toned down in the absence of evidence that increasing serine/glycine levels (through diet, for example) in vivo, OR PHGDH overexpression in T cells, improves immunotherapy outcome/tumor killing. The authors can only conclude that S/G are required for an efficient T cell response (as they do at the end of this section), not limiting.

Response:

We agree that the phrasing should be refined to better align with the presented data and to avoid overinterpreting the evidence. In response to this suggestion, we have rephrased the statement to reflect that serine and glycine are required for efficient T cell cytotoxic function rather than framing them as limiting factors. For example, the original sentence now reads: “This evidence supports the notion that serine and glycine are required for an efficient T cell cytotoxic response.”

• The serine synthesis tracing data is not interpretable in co-culture as intra/extracellular serine pools are rapidly exchanged, so the labeled serine may have come from synthesis by the cancer cells, export by the cancer cells, and uptake by the T cells, rather than synthesis from glucose. So this: “...T cells exclusively activated serine and glycine synthesis from glucose” cannot be concluded. There is not a perfect way to address this but alternate explanations should be discussed. One possibility to support their conclusions, similar to the logic employed by Faubert et al Cell 2017 (but opposite effect), is to show upstream labeling pattern of metabolites in between glucose and serine to show serine substrate labeling also increases similarly.

Response:

We greatly appreciate your comment. We agree that the potential rapid exchange of serine between cancer cells and T cells poses a challenge to conclusively attributing labeled serine in T cells to de novo synthesis from glucose. To address this, we have revised the text as follows: “Conversely, glucose-dependent labeling of serine and glycine in E0771-OVA cells

slightly decreased upon heterotypic interaction with OT-1 CD8⁺ T cells (Fig. 4e-f), suggesting that upon antigen recognition, T cells activate serine and glycine synthesis from glucose.”

To improve clarity and coherence, we analyzed the expression of PHGDH, PSAT1, and SHMT1/2 in T cells following co-culture with cancer cells. These enzymes are critical components of the serine biosynthesis pathway and one-carbon metabolism. Our results show upregulation of these genes in the T cell fraction of the co-cultures, while their expression remained unchanged in the cancer cells. This gene expression analysis complements our existing data on labeled serine and glycine incorporation, providing mechanistic support for our conclusions. To enhance the logical flow of the manuscript, we have relocated this analysis from Figure 5 to **Figures 4g-h** and **Supplementary Figures 7a-b**.

In addition, as you suggested, we explored the feasibility of examining upstream metabolite labeling patterns between glucose and serine (e.g., 3-phosphoglycerate and phosphoserine) to strengthen our conclusions. However, these metabolites could not be unambiguously detected in our datasets.

To ensure a balanced interpretation, we have expanded the discussion to explicitly acknowledge this limitation. Specifically, we agree that the labeled serine detected in T cells may originate from extracellular sources, such as export by cancer cells, rather than exclusively through glucose-driven synthesis. This alternative explanation offers a broader perspective on the metabolic interplay within the tumor microenvironment (TME).

Finally, we have emphasized that our findings underscore a metabolic dependency on serine and glycine in T cells during immune checkpoint blockade therapy, regardless of their precise source. We believe this nuanced interpretation aligns with your recommendation to consider alternative explanations while preserving the core conclusions of our study.

• The authors refer to measurements of tRNA charging throughout. However, uncharged tRNA is also present in the ribosome, particularly during starvation, and as I understand it, would also be detected in their measurement. It would thus be most accurate to call it a tRNA-ribosome association measurement. This is not necessary but they could explore total tRNA levels in their input lysate before ribosome pulldown to assess whether these effects are driven by any tRNA abundance changes. This may be relevant as Banh et al 2020 found serine limitation led to total serine tRNA level differences but no strong charging phenotype, and Hsu et al 2023 found arginine limitation reduced arginine tRNA levels.

Response:

We appreciate the reviewer’s insightful comment regarding our measurements of tRNA charging and their suggestion to more accurately describe these as “tRNA-ribosome association measurements.” We agree that uncharged tRNAs may also associate with ribosomes, particularly under starvation conditions, and thus could contribute to our observed data.

To address this, we have revised the manuscript text to clarify that our measurements reflect tRNA-ribosome association rather than exclusively tRNA charging. Specifically, we now use the term “tRNA-ribosome association measurements” throughout the manuscript to provide a more precise interpretation of our findings.

We also thank the reviewer for suggesting that we assess total tRNA levels in the input lysates prior to ribosome pulldown to determine whether changes in tRNA abundance could be driving the observed effects. While our current study is focused on ribosome-associated tRNAs, we acknowledge the importance of distinguishing between changes in total tRNA abundance and ribosome-associated tRNA levels. We have added a discussion of this point to the manuscript

and cited relevant studies (e.g., Banh et al., 2020) that highlight how nutrient limitations can influence total tRNA levels.

We emphasize that our conclusions are based solely on ribosome-associated tRNA measurements, and we do not claim to infer changes in total tRNA levels. However, we recognize that future analyses of total tRNA abundance could provide valuable complementary insights and further clarify the relationship between nutrient availability, tRNA abundance, and ribosome association.

• PC1 has to explain a greater proportion of the variance than PC2. This needs to be addressed. It also looks like something is off because the points are similarly separated across PC1 and 2 but they account for vastly different amounts of the variance in the samples.

Response:

Thank you for pointing this out. The values for PC1 and PC2 variance were rechecked, and indeed there was a mislabeling of the figure, this has been corrected in the revised version.

• Fig2e should be displayed as points (ideally an entire gene set) highlighted on a volcano plot to clarify how these differences compare to everything else changing.

Response:

Thank you for your suggestion. In the revised manuscript (Extended Data Fig. 4d–e), we have included volcano plots that display all lysosomal genes, both unchanged and differentially expressed, as discussed in the study. The plots clearly define thresholds for fold change and statistical significance. Our analysis underscores that lysosomal genes are among the most significantly overexpressed in MRC5 and SUM-159PT cells following heterotypic interactions.

• “DQ BSA fluorescence and ribosome association of aminoacylated Ala- and Gly-tRNAs were strongly decreased in LYSET-deficient cells upon co-culture with 186 MRC5 cells (Fig. 2f-h;...” – typo, should say Fig 3f-h.

Response:

Thank you for catching that error and bringing it to our attention. You are absolutely correct; the text should reference Fig. 3f-h, not Fig. 2f-h. We made the necessary correction in the revised manuscript to ensure accuracy and clarity for readers.

• The authors find serine codons are equally affected by serine/glycine limitation. This seems to conflict with recent work by Bahn et al 2020 showing that only specific codons (TCC/TCT) are affected by serine limitation. Can the authors discuss this?

Response:

Thank you for pointing out this apparent discrepancy. We agree that it is an important topic for discussion. Our findings indicate that all serine codons are affected under the conditions we studied. In the revised version of our manuscript, we address the results of Bahn et al. (2020), highlighting key differences in experimental design—such as the duration of deprivation and the cell types used—and discuss the potential implications for codon-specific effects during serine deprivation.

References.

1. Loayza-Puch, F., Rooijers, K., Buil, L.C.M., Zijlstra, J., Oude Vrielink, J.F., Lopes, R., Ugalde, A.P., van Breugel, P., Hofland, I., Wesseling, J., et al. (2016). Tumour-specific proline vulnerability uncovered by differential ribosome codon reading. *Nature* 530, 490–494.
2. Possemato, R., Marks, K.M., Shaul, Y.D., Pacold, M.E., Kim, D., Birsoy, K., Sethumadhavan, S., Woo, H.-K., Jang, H.G., Jha, A.K., et al. (2011). Functional genomics reveal that the serine synthesis pathway is essential in breast cancer. *Nature* 476, 346–350.
3. Geeraerts, S.L., Heylen, E., De Keersmaecker, K., and Kampen, K.R. (2021). The ins and outs of serine and glycine metabolism in cancer. *Nat Metab* 3, 131–141.

Response to Reviewers' Comments

We sincerely appreciate the reviewers' thorough evaluation of our manuscript and their constructive feedback, which has significantly improved the clarity and rigor of our study. We have carefully addressed each of the comments and have made the necessary revisions to the text, figures, and supplementary materials accordingly. Below, we provide detailed responses to each point raised by the reviewers, outlining the changes made and clarifying aspects of our analysis where needed. We believe these revisions enhance the manuscript and hope that it is now suitable for publication.

Reviewer #2 (Remarks to the Author):

Most of my comments have been addressed and the figures and discussion have been amended accordingly. However, some points still need to be detailed/addressed:

- In the transwell setup, have the OTI cells been activated with aCD3/aCD28 (revised Supp Fig 6l)? Please add this in the figure legend and add the details of the transwell experiment in the material and method section.

Response:

Thank you for your comment. Yes, in the transwell setup, the OTI cells were activated with aCD3/aCD28. We have now explicitly stated this in the revised legend for Supplementary Figure 6l. Additionally, we have included a detailed description of the transwell experiment in the Materials and Methods section to ensure clarity and reproducibility.

- One of my comments was to describe the impact of both ser/gly deprivation and α -PD1 treatment on T cell functions. The authors provided answers for T cell infiltration and proliferation only. Are there any limitations in their model that prevent the authors from analyzing T cell functions (cytokine and cytotoxic molecule production) when isolated from the tumor (revised Supp Fig 8f)?

Response:

We thank the reviewer for their feedback. Regarding the assessment of T cell functions, including cytokine and cytotoxic molecule production, certain technical limitations restricted these analyses. Specifically, the number of tumor-infiltrating T cells recovered after serine/glycine deprivation and α -PD1 treatment was insufficient for reliable functional assays. Additionally, the metabolic stress induced by serine/glycine deprivation may have affected T cell viability and cytokine production, potentially confounding the results. To address this limitation, we have carefully revised our conclusions and have avoided making definitive statements regarding T cell functionality in tumors in response to serine/glycine deprivation and α -PD1 treatment. Furthermore, while these analyses were conducted *in vitro*, we believe the data presented in Figures 4d, 5d, and Supplementary Figures 4d–e and 6h sufficiently demonstrate the impact of serine/glycine deprivation on T cell function. We appreciate the reviewer's suggestion and believe that these revisions enhance our study by providing a more comprehensive evaluation of T cell responses.

Reviewer #3 (Remarks to the Author):

The authors have addressed my major concerns excepting the following points on which I still disagree. I would advocate for minor changes to address these remaining points before publication, which I otherwise support.

1. *“Regarding the dip observed at position -12 in the ribosome density plots, this feature reflects a reduced ribosome density at the P site of the codon, which aligns with transient*

stalling of ribosomes at the A site (position -15).” I don’t follow the logic/interpretation here and I don’t think this should be labeled as a fact in the legend, particularly without indicating the P/E sites on the plot. I think the authors are saying that because there is more ribosome density where that codon is exactly in the A-site, there is necessarily less where that codon is in the P-site. Why would this not apply to the E-site also? Also, their traces don’t appear to be this precise – for the Ala codons, the dip doesn’t occur, and for Ser codons, the dip occurs at position 0. This P-site dip is also not observed in other papers that show significant A-site specific ribosome stalling (see Ishimura et al 2016). I am honestly not convinced regarding the Diricore analysis that this peak is not in the same realm as the technical noise. However, the authors follow up data, particularly the ribosome-tRNA association, is interesting and valid and supports their ideas enough to let the Diricore stand – though I would consider acknowledging the limitations of this method with respect to signal to noise to avoid misleading others. There has been great debate and optimization in the ribosome profiling field about how to accurately capture ribosome positions with high resolution at codon resolution, and what artifacts might be inevitable due to library or harvesting methods, so I think the authors would really benefit from care here.

Response:

We appreciate the reviewer’s comments regarding the interpretation of the dip observed at position -12 in the ribosome density plots. In response, we have removed the corresponding text from the figure legends to ensure that interpretations are not overstated. Additionally, we have expanded our discussion to acknowledge the limitations of the Diricore method, particularly in distinguishing biologically relevant ribosome occupancy from technical noise. We recognize that ribosome profiling datasets require careful interpretation due to inherent variability in ribosome positioning, and we have now explicitly discussed potential artifacts that may arise due to library preparation or harvesting methods. We hope these revisions clarify our analysis and provide a more balanced perspective on the Diricore findings.

2. “Combining both treatments resulted in a robust increase in lysosomal activity in mono-cultures, as shown in Extended Data Fig. 4e.” – I think they mean Ex Data Fig 4g, and I agree this data addresses my concern.

Response:

Thank you for pointing out the figure reference. We appreciate the clarification. We are also glad to hear that the data effectively address your concern.

3. “TNBC cells lacking PHGDH show a heightened dependency on extracellular serine. In these cells, serine is not only taken up from the environment but is also synthesized from glycine through the action of serine hydroxymethyltransferase (SHMT)3 . This conversion process becomes crucial when de novo synthesis from glucose is compromised.” This is an interesting point and would be easy to test by showing that PHGDH overexpression rescues Ala and Gly stalling upon low glucose. Not within the scope of the current study, I agree. “Consistent with our Diricore analysis, we observed a significant decrease in the intracellular concentrations of alanine and glycine.” I remain skeptical that a ~2-fold decrease in intracellular alanine and glycine pools is enough to limit protein synthesis (as the authors observe with proline). Do the authors propose that alanine and glycine tRNA charging has a significantly higher KM than proline tRNA charging? This data should definitely be included in the manuscript and discussed for transparency.

Response:

We appreciate the reviewer’s suggestion regarding PHGDH overexpression as a potential rescue experiment. While this is beyond the scope of our current study, we agree that it represents an intriguing avenue for future research to further elucidate the metabolic dependencies of TNBC cells.

With respect to intracellular alanine and glycine levels, we emphasize that the changes we observed are relative rather than absolute concentrations. Notably, intracellular proline levels are considerably higher than those of glycine and alanine, which may influence tRNA aminoacylation dynamics (**Figure R1**). To directly assess the impact of these metabolic shifts on tRNA charging, we performed tRNA aminoacylation assays in SUM-159PT cells cultured under glucose-limiting conditions for 24 and 48 hours. Our results show that reductions in alanine and glycine specifically impair Ala- and Gly-tRNA charging (**Supplementary Fig. 3I**), whereas Pro-tRNA charging remains unaffected.

Figure R1. SUM-159PT were grown in full medium (Control) or in limiting concentrations of glucose (2,5 mM, Glucose (-)). Amino acids were measured by LC-MS/MS. Data represent mean \pm SD (n = 3); *** P < 0.01 by Student's t-test.

We have incorporated these data, along with our metabolomics findings, into the manuscript (**Supplementary Fig. 3k and 3I**) to address potential differences in tRNA charging kinetics among alanine, glycine, and proline. We believe these additions enhance transparency and provide a more comprehensive context for our conclusions.

4. I appreciate all of the changes the authors made to phrasing to not oversell the point that these amino acids are limiting protein production, that serine is limiting for the T cell response, and that they are measuring a tRNA-ribosome association and not a charged tRNA-ribosome association.

Response:

We appreciate the reviewer's comment and are grateful for the constructive feedback that helped us improve the manuscript.